# Loss of a proteostatic checkpoint in intestinal stem cells contributes to age-related epithelial dysfunction

Imilce A. Rodriguez-Fernandez[1,2], Yanyan Qi[1] & Heinrich Jasper[1,2,3]

A decline in protein homeostasis (proteostasis) has been proposed as a hallmark of aging. Somatic stem cells (SCs) uniquely maintain their proteostatic capacity through mechanisms that remain incompletely understood. Here, we describe and characterize a 'proteostatic checkpoint' in *Drosophila* intestinal SCs (ISCs). Following a breakdown of proteostasis, ISCs coordinate cell cycle arrest with protein aggregate clearance by Atg8-mediated activation of the Nrf2-like transcription factor cap-n-collar C (CncC). CncC induces the cell cycle inhibitor Dacapo and proteolytic genes. The capacity to engage this checkpoint is lost in ISCs from aging flies, and we show that it can be restored by treating flies with an Nrf2 activator, or by over-expression of CncC or Atg8a. This limits age-related intestinal barrier dysfunction and can result in lifespan extension. Our findings identify a new mechanism by which somatic SCs preserve proteostasis, and highlight potential intervention strategies to maintain regenerative homeostasis.

[1] Buck Institute for Research on Aging, 8001 Redwood Boulevard, Novato, CA 94945-1400, USA. [2] Immunology Discovery, Genentech, Inc., 1 DNA Way, South San Francisco, California 94080, USA. [3] Leibniz Institute on Aging - Fritz Lipmann Institute, Jena 07745, Germany. Correspondence and requests for materials should be addressed to H.J. (email: jasperh@gene.com)

Protein Homeostasis (Proteostasis) encompasses the balance between protein synthesis, folding, re-folding and degradation, and is essential for the long-term preservation of cell and tissue function. It is achieved and regulated by a network of biological pathways that coordinate protein synthesis with degradation and cellular folding capacity in changing environmental conditions[1]. This balance is perturbed in aging systems, likely as a consequence of elevated oxidative and metabolic stress, changes in protein turnover rates, decline in the protein degradation machinery, and changes in proteostatic control mechanisms[2–5]. The resulting accumulation of misfolded and aggregated proteins is widely observed in aging tissues, and is characteristic of age-related diseases like Alzheimer's and Parkinson's disease. The age-related decline in proteostasis is especially pertinent in long-lived differentiated cells, which have to balance the turnover and production of long-lived aggregation-prone proteins over a timespan of years or decades. But it also affects the biology of somatic stem cells (SCs), whose unique quality-control mechanisms to preserve proteostasis are important for stemness and pluripotency[6,7].

Common mechanisms to surveil, protect from, and respond to proteotoxic stress are the heat shock response (HSR) and the organelle-specific unfolded protein response (UPR). When activated, both stress pathways lead to the upregulation of molecular chaperones that are critical for the refolding of damaged proteins and for avoiding the accumulation of toxic aggregates. If changes to the proteome are irreversible, misfolded proteins are degraded by the proteasome or by autophagy[6,8]. While all cells are capable of activating these stress response pathways, SCs deal with proteotoxic stress in a specific and state-dependent manner[6]. Embryonic SCs (ESCs) exhibit a unique pattern of chaperone expression and elevated 19S proteasome activity, characteristics that decline upon differentiation[9–11]. ESCs share elevated expression of specific chaperones (e.g. HspA5, HspA8) and co-chaperones (e.g., Hop) with mesenchymal SCs (MSCs) and neuronal SCs (NSCs)[12], and elevated macroautophagy (hereafter referred to as autophagy) with hematopoietic SCs (HSCs), MSCs, dermal, and epidermal SCs[6,13]. Defective autophagy contributes to HSC aging[14]. It has further been proposed that SCs can resolve proteostatic stress by asymmetric segregation of damaged proteins, a concept first described in yeast[15–18].

While these studies reveal unique proteostatic capacity and regulation in SCs, how the proteostatic machinery is linked to SC activity and regenerative capacity, and how specific proteostatic mechanisms in somatic SCs ensure that tissue homeostasis is preserved in the long term, remains to be established. Drosophila intestinal stem cells (ISCs) are an excellent model system to address these questions. ISCs constitute the vast majority of mitotically competent cells in the intestinal epithelium of the fly, regenerating all differentiated cell types in response to tissue damage. Advances made by numerous groups have uncovered many of the signaling pathways regulating ISC proliferation and self-renewal[19]. In aging flies, the intestinal epithelium becomes dysfunctional, exhibiting hyperplasia and mis-differentiation of ISCs and daughter cells[20]. This age-related loss of homeostasis is associated with inflammatory conditions that are characterized by commensal dysbiosis, chronic innate immune activation, and increased oxidative stress[21–23]. It further seems to be associated with a loss of proteostatic capacity in ISCs, as illustrated by the constitutive activation of the unfolded protein response of the endoplasmic reticulum (UPR-ER), which results in elevated oxidative stress, and constitutive activation of JNK and PERK kinases[24,25]. Accordingly, reducing PERK expression in ISCs is sufficient to promote homeostasis and extend lifespan[25].

ISCs of old flies also exhibit chronic inactivation of the Nrf2 homologue CncC[26]. CncC and Nrf2 are considered master regulators of the antioxidant response, and are negatively regulated by the ubiquitin ligase Keap1. In both flies and mice, this pathway controls SC proliferation and epithelial homeostasis[26,27]. It is regulated in a complex and cell-type specific manner[26,28,29]. Canonically, Nrf2 dissociates from Keap1 in response to oxidative stress and accumulates in the nucleus, inducing the expression of antioxidant genes[28]. Drosophila ISCs, in turn, exhibit a 'reverse stress response' that results in CncC inactivation in response to oxidative stress. This response is required for stress-induced ISC proliferation, including in response to excessive ER stress, and is likely mediated by a JNK/Fos/Keap1 pathway[24,26] (Li, Hochmuth, Jasper, unpublished).

The Nrf2 pathway has also been linked to proteostatic control: 'Non-canonical' activation of Nrf2 by proteostatic stress as a consequence of an association between Keap1 and the autophagy scaffold protein p62 has been described in mammals[30–35]. A similar non-canonical activation of Nrf2 has been described in Drosophila, where CncC activation is a consequence of the interaction of Keap1 with Atg8a, the fly homologue of the autophagy protein LC3[36]. Nrf2/CncC activation induces proteostatic gene expression, including of p62 in mammalian cells[32] and of p62/Ref[2]P and LC3/Atg8a in flies[36]. Nrf2 is further a central transcriptional regulator of the proteasome in both Drosophila and mammals[37–39]. Whether and how Nrf2 also influences proteostatic gene expression in somatic SCs remains unclear.

Here, we show that Drosophila CncC links cell cycle control with proteostatic responses in ISCs via the accumulation of dacapo, a p21 cell cycle inhibitor homologue, as well as the transcriptional activation of genes encoding proteases and proteasome subunits. We establish that this program constitutes a transient 'proteostatic checkpoint', which allows clearance of protein aggregates before cell cycle activity is resumed. In old flies, this checkpoint is impaired and can be reactivated with a CncC activator.

## Results

**A proteostatic checkpoint in ISCs.** To explore the consequences of proteostatic stress for ISC function, we knocked down various components of the proteasome, including Rpn3, a 19S lid subunit, as well as Rpn2 and Rpt6R, 19S base subunits of the proteasome, by RNAi in ISCs[40], and traced the ISC lineage using a Flp-Out strategy[41,42]. As expected, this resulted in an accumulation of poly-ubiquitinated protein aggregates in ISCs and their daughter cells (Fig. 1a, Supplementary Fig. 1a). At the same time, we observed that the size of ISC-derived lineages was much smaller than those derived from wild-type control cells (Fig. 1b, Supplementary Fig. 1a).

We reasoned that this reduction in clone growth could be a consequence of (i) unspecifically impaired ISC function due to proteostatic stress, (ii) accumulation of cell cycle regulators due to decreased proteasome function, (iii) increased cell death of differentiated cells in such conditions, or (iv) a specific cell cycle arrest in ISCs due to the presence of protein aggregates. To distinguish between these possibilities, we tested the proliferative response of ISCs to protein aggregates that formed by an aggregation-prone protein directly. We performed a time-course experiment in which a fluorescently labeled and aggregation-prone version of human Huntingtin (mRFP-Htt$^{Q138}$)[43] was expressed in ISCs for a short period of time, followed by a chase period in which the transgene was not transcribed. Short-term expression of mRFP-Htt$^{Q138}$ allowed monitoring the fate of RFP-labeled protein aggregates at various time points after expression of the transgene, and avoided cell death reported when cells constitutively express polyQ proteins[44]. We used the TARGET system, in which temperature-controlled expression of UAS-

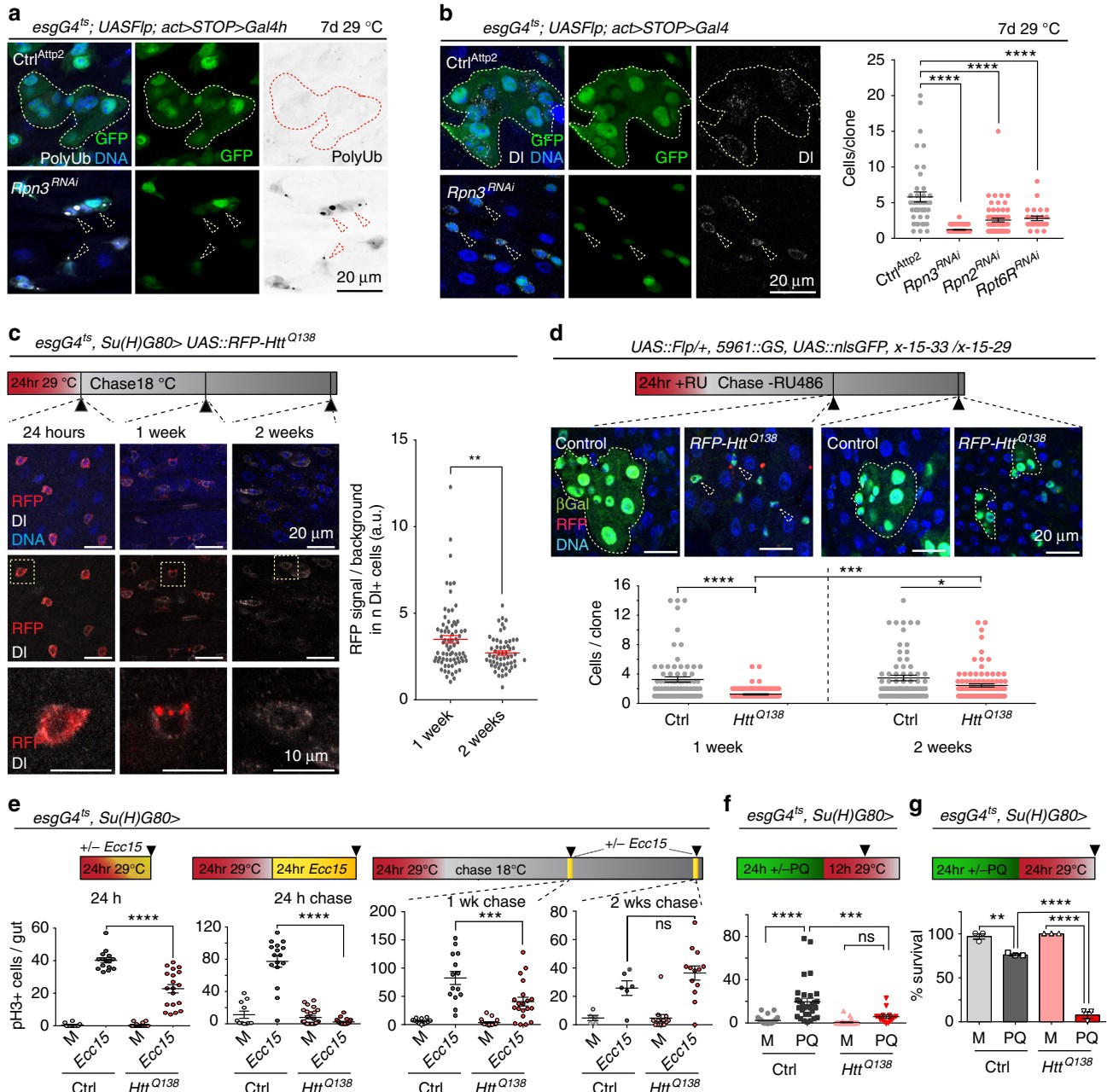

**Fig. 1** Protein aggregates induce a transient cell cycle arrest in ISCs. **a**, **b** GFP-marked lineages (green) from *Rpn3^RNAi* expressing ISCs in 8-day-old flies. Immunohistochemistry: **a** anti-poly-ubiquitin (white, left; black, right), **b** anti-Delta (white). **b** 1-way ANOVA / Dunnett's (*n* = 41, 73, 66, and 25 clones from 3, 4, 4, 3 independent animals respectively). **c** *mRFP-Htt^Q138* expressed for 24 h in ISCs of 11-day-old flies. Delta (Dl; white). Graph: RFP signal normalized to background. Two-tailed unpaired t-test: (*n* = 75 and 57 Dl+ cells from five and three animals respectively, from two independent experiments). **d** ISC lineages from 4 days old flies labelled with β-galactosidase (βgal, green) using Flp-mediated somatic recombination of a split lacZ gene (X-15-33/x-15-29[47]), carrying or not *mRFP-Htt^Q138* (RU486-inducible ISC/EB driver 5961::GS). Beta-galactosidase (green), RFP-labeled aggregates red. Yellow arrowheads point out βgal-labeled ISCs. 1-way ANOVA/Sidak's (*n* = 76, 154, 72, and 97 clones from 10, 13, 9, and 9 animals respectively, from two independent experiments). **a**–**d**, Representative areas of posterior midgut. DNA (Hoechst, blue); scale bars 20 μm and 10 μm (**c**, inset). **e** Quantification of phospho-Histone H3 (pH3)+ cells after 24 h expression of *mRFP-Htt^Q138* in ISCs of 6 to 8-day-old flies during infection with *Ecc15* (or Mock, M; first graph), or followed by 24 h *Ecc15* immediately (second graph), or by 8 h *Ecc15* one week (third graph), or two weeks (fourth graph) later. Two-way ANOVA with Tukey's (*n* = 12, 15, 20, 19 guts (first); 11, 16, 26, 24 guts (second); 12, 14, 19, 20 guts (third); 5, 6, 13, 13 guts (fourth graph)). **f** pH3+ cell quantification in 12-day-old flies after 5 mM Paraquat (PQ) or Mock (M) treatment at 25 °C, followed by regular food at 29 °C for 12 h. Two-way ANOVA with Sidak's (*n* = 32, 32, 40, 15 guts, from two independent experiments). **g** Survival of 12-day-old flies fed 5 mM PQ or Mock at 25 °C for 24 h, followed by regular food at 29 °C for 24 h (*n* = 3 biological replicates). Two-way ANOVA with Tukey's. Means and s.e.m. shown in all graphs; ns, not significant, ****$P < 0.0001$, ***$P < 0.001$, **$P < 0.01$, *$P < 0.05$. Genotypes in Supplementary Table 1

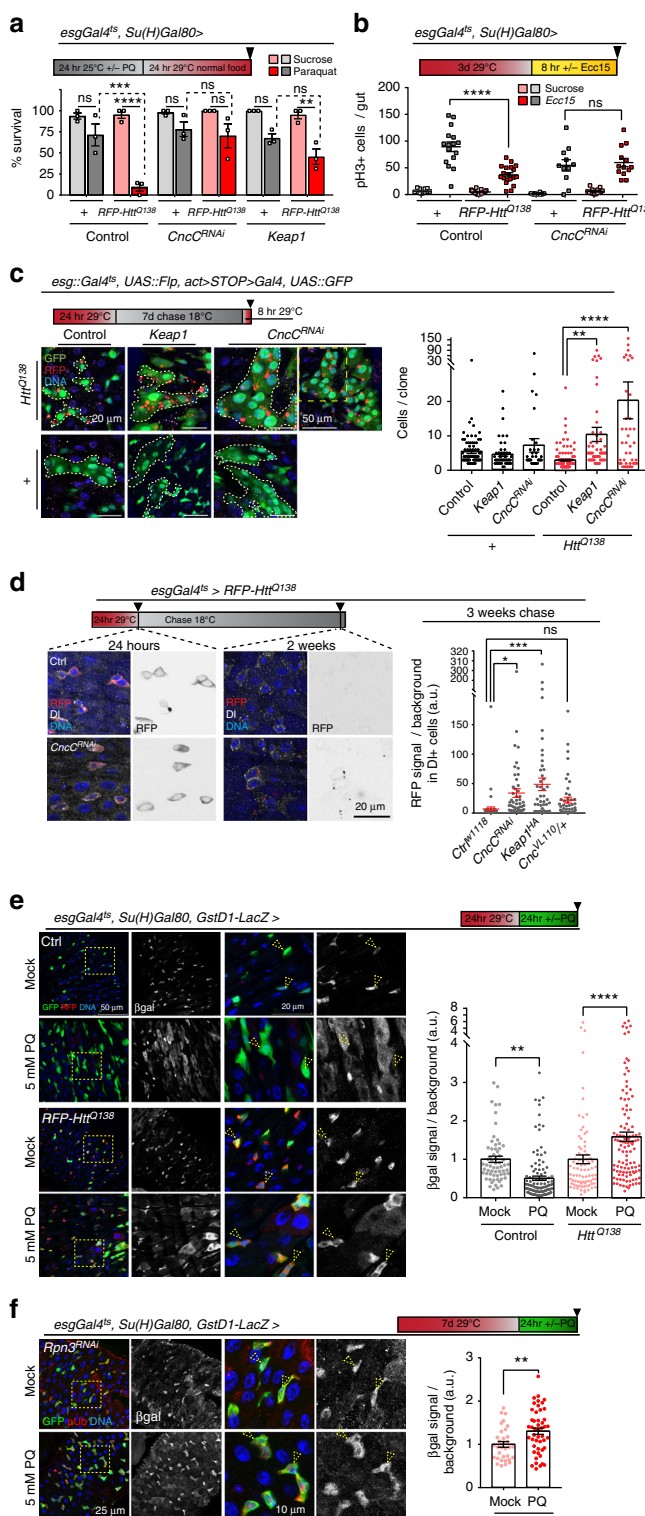

**Fig. 2** Impaired proteostatic checkpoint in CncC-deficient ISCs. **a** Survival of cohorts with indicated genotypes after PQ exposure. Eleven-day-old flies fed 5 mM PQ in 5% sucrose at 25 °C for 24 h, then shifted to regular food at 29 °C for 24 h. Percent survivors after 24 h (three technical replicates from two biological replicates; two-way ANOVA with Tukey's test). **b** Expression of $mRFP\text{-}Htt^{Q138}$ and $CncC^{RNAi}$ in ISCs of 6-day-old flies for 3 days followed by 8 h of $Ecc15$ infection. anti-pH3 labels mitotic ISCs. 1-way ANOVA with Tukey's ($n = 14, 16, 19, 19, 14, 13, 10$, and 11 animals, respectively). **c** ISC-derived lineages (green, outlined with yellow lines) in 7-day-old flies co-expressing $mRFP\text{-}Htt^{Q138}$ and $CncC^{RNAi}$ or Keap1. Top right panel shows lower magnification of a large clone expressing $CncC^{RNAi}$ ($mRFP\text{-}Htt^{Q138}$ in red). One-way ANOVA with Sidak's ($n = 80, 53, 39, 153, 61, 41$ clones from 8, 5, 5, 8, 7, 8 animals, respectively). **d** Expression of $mRFP\text{-}Htt^{Q138}$ alone or combined with $CncC^{RNAi}$, Keap1HA or $CncC^{VL110}/+$ in ISCs and EBs of 11-day-old flies for 24 h. Presence of aggregates (red/black) assessed immediately (24 h), or 2 and 3 weeks after expression. Dl staining was used to identify ISCs. Graph: RFP signal in Dl+ cells normalized to background shown ($n = 53, 49, 52, 49$ Dl+ cells from 7, 5, 7, 6 animals, respectively, from two independent experiments). One-way ANOVA with Dunnett's. **e**, **f** 12-day-old flies carrying the gstD1::LacZ reporter expressing (**e**) $mRFP\text{-}Htt^{Q138}$ or (**f**) $Rpn3^{RNAi}$ transgene for 24 h at 29 °C followed treatment with (**e**) 5 mM or (**f**) 7.5 mM PQ in 5% sucrose or 5% sucrose alone (mock). βgalactosidase (βgal), white; GFP, green; yellow arrowheads indicate selected ISCs. **e** $mRFP\text{-}Htt^{Q138}$ or **f** poly-Ubiquitin (pUb) aggregates in red. **e** One-way ANOVA with Sidak's ($n = 63, 122, 84, 112$ GFP+ cells from 4, 7, 5, 6 animals, respectively). **f** Unpaired student t-test ($n = 33, 48$ GFP+ cells from 4 and 5 animals, respectively). Higher magnification shown in right panels. **c–f** Representative areas in posterior midgut, DNA (Hoechst) in blue, and timeline indicated in figure. Scale bars: 20 μm for (**c**, **d**; **e**, insets); 50 μm for (**c**, insets); **e**); 25 μm for (**f**); 10 μm (**f**, inset). Means and s.e.m. shown in all graphs; ns, not significant, ****$P < 0.0001$, ***$P < 0.001$, **$P < 0.01$. Genotypes in Supplementary Table 1

became the only observable RFP-positive structures in the ISC cytoplasm, and were still present 1 week after the pulse, yet disappeared after 2 weeks (Fig. 1c). To obtain a quantitative metric for the presence of these puncta in ISCs, we measured RFP signal intensity, as puncta were heterogenous in size and number in individual ISCs. Since after about 48 h only localized puncta and no background fluorescence were detected in ISCs, this quantification represented the presence of RFP-labeled protein aggregates in ISCs at the 1 week and 2 week timepoints after induction (Fig. 1c).

Lineage-tracing experiments revealed that the loss of visible RFP puncta is not due to replacement of aggregate-bearing ISCs, but due to elimination of RFP+ aggregates in ISCs that are present at the time of induction: We used a split-lacZ strategy to induce marked ISC lineages in response to the drug RU486 in the background of $mRFP\text{-}Htt^{Q138}$ expression[47] (Fig. 1d, Supplementary Fig. 1b). These lineage-tracing experiments also demonstrated that during the period of $mRFP\text{-}Htt^{Q138}$ clearance, ISCs have a transiently reduced division rate compared to wild-type controls, but recover proliferative activity after $mRFP\text{-}Htt^{Q138}$ puncta are eliminated 2 weeks after induction (Fig. 1d, Supplementary Fig. 1b). This observation is consistent with the engagement of a transient cell-cycle arrest in ISCs bearing protein aggregates. Importantly, the number of observable cell clones did not change in the course of the experiment, supporting the notion that $mRFP\text{-}Htt^{Q138}$ expressing ISCs are not damaged and do not get eliminated, but reduce their proliferation rate transiently before returning to their normal proliferation state (Supplementary Fig. 1c). The introduction of aggregation-prone proteins in cells of the wing imaginal disc did not result in slower

linked transgenes is achieved by co-expression of Gal4 with a temperature-sensitive version of the Gal4 inhibitor Gal80[45,46]. Expression of $mRFP\text{-}Htt^{Q138}$ was induced for 24 h, and was followed by a chase period of one or two weeks, upon which the presence of aggregates in ISCs was evaluated by confocal microscopy (Fig. 1c). While initially homogeneous, the RFP signal in ISCs progressively concentrated into brightly labeled puncta after expression of mRFP-Htt$^{Q138}$, indicating the formation of aggregates that were observed in the ISC cytoplasm starting at about 1 day after induction. These puncta eventually

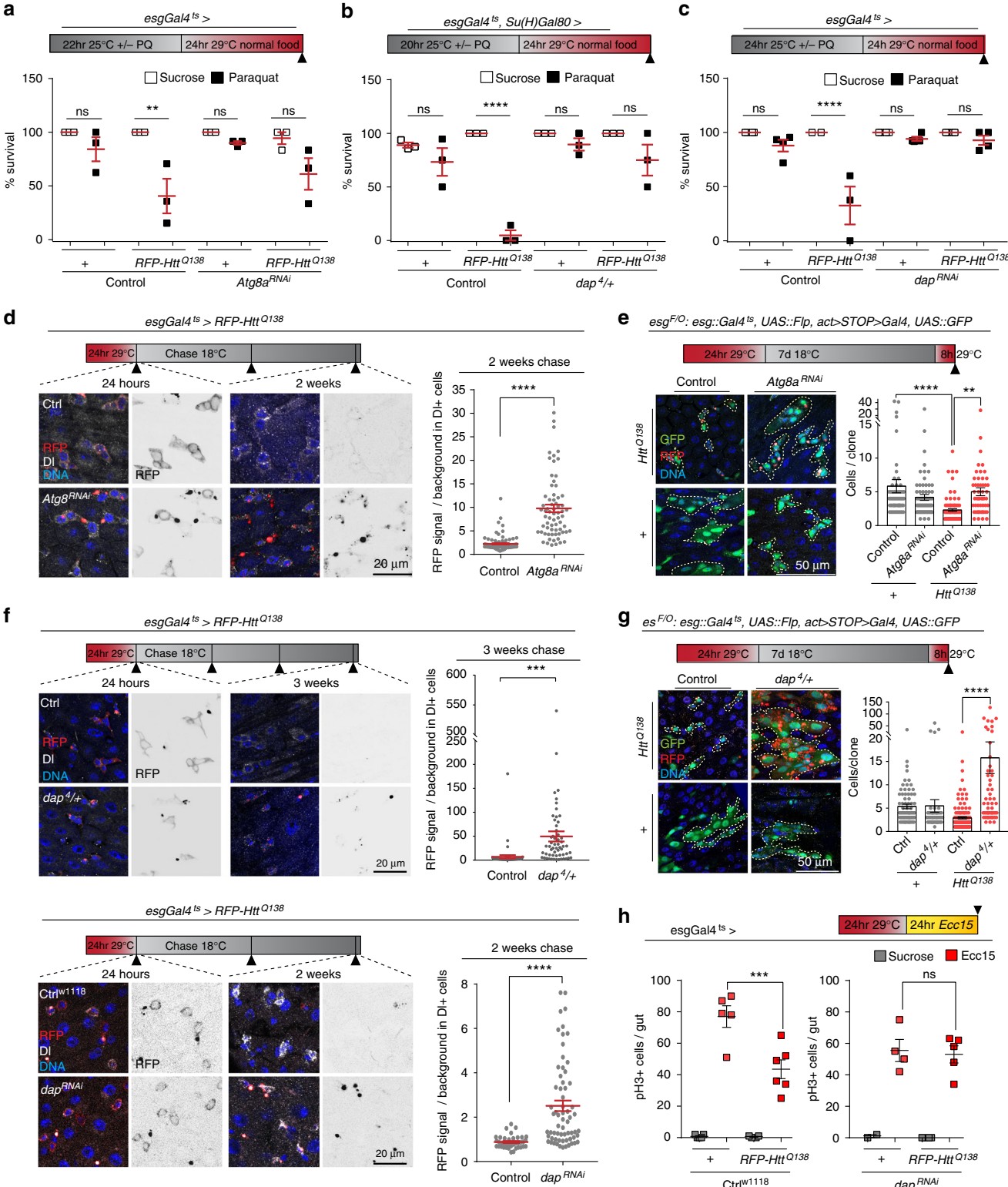

growth of cell clones, indicating that the observed reduction in cell proliferation may be a unique response of ISCs (Supplementary Fig. 1c).

To resolve the timing of proliferative impairment by protein aggregates in ISCs more precisely, we assessed whether and how rapidly *mRFP-Htt[Q138]* expression can inhibit ISC proliferation when induced by enteropathogen infection. In wild-type flies, infection with *Erwinia carotovora carotovora 15 (Ecc15)* leads to a transient but vigorous proliferative response of ISCs that can be

carefully timed[48,49]. We detected a slight but significant inhibition of proliferation when *mRFP-Htt[Q138]* was expressed at the same time as flies were exposed to *Ecc15* (Fig. 1e), and a stronger inhibition when *mRFP-Htt[Q138]* was expressed 24 h before *Ecc15* infection (Fig. 1e). This inhibition was still observed one week after *mRFP-Htt[Q138]* expression (Fig. 1e; Supplementary Fig. 1d), and is a cell-autonomous consequence of *mRFP-Htt[Q138]* expression in ISCs, since expressing Htt[Q138] in enteroblasts (EBs) did not impair the proliferative response to infection (Supplementary

**Fig. 3** *Atg8a* and *dacapo* regulate the proteostatic checkpoint. **a–c** PQ-induced lethality in 13–14-day-old flies of indicated genotypes. **a** 7.5 mM PQ, 14-day-old flies; three technical, two biological replicates; **b** 5 mM PQ, 13-day-old flies; three technical, two biological replicates; **c** 7.5 mM PQ, 14-day-old flies; two biological, 2–4 technical replicates per sample. 2-way ANOVA with Sidak's. **d, f** Images of posterior midguts of flies of the indicated genotypes after the indicated interventions and times. Seven-day (**d**) or 11-day (**f**) old flies. Aggregates in red or black, Dl in white. Graphs: RFP signal in Dl+ cells normalized to background ($n = 100$, 68 from seven and eight animals respectively in (**d**); $n = 53$, 57 from seven and eight animals respectively in (**f**), top panel, experiments performed in parallel with Fig. 2d, controls are the same; $n = 38$, 65 from four and eight animals respectively in (**f**), bottom panel). Unpaired two-tailed *t*-test. **e** ISC-derived lineages (green, outlined yellow) of 10-day-old flies of indicated genotypes. Aggregates in red. Graph: Number of cells per clone ($n = 61$, 75, 85, 52 from 7, 6, 5, and 7 animals, respectively). **g** ISC-derived Flp-out clones (green, yellow outline) of indicated genotypes. Number of cells per clone ($n = 80$, 54, 153, 55 from 8, 8, 8, and 11 animals, respectively). **e** and **g**: 1-way ANOVA with Tukey's, experiments performed in parallel with Fig. 2e, controls are the same. **d–g** Representative areas in posterior midgut, DNA (Hoechst, blue). Scale bars: 20 μm (**d, f**); 50 μm (**e, g**). **h** Quantification of pH3+ cells in guts from 10-day-old flies with indicated genotypes and treatment. 1-way ANOVA with Sidak's ($n = 5$, 5, 4, 6, 2, 4, 4, 5 guts from independent animals). Means and s.e.m. shown in all graphs; ns not significant, ****$P < 0.0001$, ***$P < 0.001$, **$P < 0.01$. Genotypes in Supplementary Table 1

Fig. 1e). If infection was performed in flies 2 weeks after *mRFP-Htt^Q138* expression (when RFP+ aggregates have been cleared), there was no significant difference in mitotic figures in guts of wild-type flies and flies expressing Htt^Q138 in ISCs (Fig. 1e; Supplementary Fig. 1d). To exclude the possibility that inhibition of proliferation in the presence of Htt^Q138 aggregates was specific to *Ecc15* infection, we induced regenerative pressure by feeding flies the oxidative stress-inducing agent Paraquat (PQ)[50]. PQ treatment induces proliferation of wild-type ISCs, but not of ISCs in which Htt^Q138 was expressed for 12 h (Fig. 1f). Performing the same experiment, but extending the period of Htt^Q138 expression to 24 h, leads to increased mortality of flies that had been treated with PQ (Fig. 1g). This increase in mortality is consistent with impaired epithelial regeneration as a consequence of reduced ISC proliferation.

Together, these observations indicate that ISCs undergo a transient cell cycle arrest in response to protein aggregate formation, that this arrest is sustained until aggregates are cleared, and that the affected ISCs can undergo normal proliferative responses after aggregate clearance. We termed this phenomenon a 'proteostatic checkpoint'.

**Nrf2/CncC regulates the proteostatic checkpoint.** To explore the molecular mechanisms by which the proteostatic checkpoint is engaged and regulated in ISCs, we used the mortality of PQ-treated animals with Htt^Q138-expressing ISCs as a readout for a limited genetic candidate screen including proteostatic and cell cycle regulators (dacapo, Cyclin E, Atg8a/LC3, and ref 2P/p62), Cul4 and an associated protein (Cul4, Dcaf12), and histone deacetylases (Sir2, Sirt2, Sirt4, Sirt7, and HDAC6). This screen included CncC and Keap1, which emerged as candidates due to their roles in proliferative control of ISCs[26] and proteostasis[36–39]. Limiting CncC activity in ISCs, either by knocking down CncC itself or by over-expressing Keap1, was sufficient to rescue the elevated Paraquat-induced mortality of Htt^Q138-expressing animals (Fig. 2a). Knocking down CncC in ISCs also prevented the inhibition of *Ecc15*-induced proliferation of ISCs after expression of Htt^Q138 (Fig. 2b; note that induction of proliferation by *Ecc15* is reduced in CncC-deficient ISCs compared to wild-type controls). These results suggested that CncC was involved in the proteostatic checkpoint, a notion that we confirmed in lineage-tracing studies using the *esg::Gal4^ts*, Flp-out system[41,42]. Clones derived from ISCs in which CncC was knocked down or in which Keap1 was over-expressed, grew at a higher rate than wild-type clones, and exhibited increased levels of *mRFP-Htt^Q138* puncta (Fig. 2c). Impaired clearance of *mRFP-Htt^Q138* puncta in CncC-deficient ISCs was further confirmed in TARGET experiments at 2 weeks after induction, a time point in which control ISCs had already cleared the *mRFP-Htt^Q138* aggregates (Fig. 2d; note that the prevalence of *mRFP-Htt^Q138* signal in ISCs was quantified

3 weeks after induction, a timepoint where wild-type cells very rarely exhibit a remaining signal). In these experiments, CncC deficiency was induced by knocking down CncC, over-expressing Keap1, or by performing the experiments in *CncC* heterozygous conditions.

Using the Nrf2 activity reporter GstD1::LacZ[29], we confirmed that Nrf2 is activated in ISCs in the presence of *mRFP-Htt^Q138* aggregates, or when Rpn3 is knocked down (Fig. 2e, f): Wild-type ISCs exhibit high basal, CncC-dependent expression of this reporter, yet in conditions of regenerative pressure, CncC is selectively inactivated in ISCs in a Keap1-dependent manner[26]. This inactivation is essential for ISC proliferation, as it allows elevation of reactive oxygen species in ISCs[26]. In ISCs bearing *mRFP-Htt^Q138* aggregates, however, repression of GstD1::lacZ expression was not observed, indicating that the presence of aggregates impairs the ability of ISCs to inactivate CncC under regenerative pressure.

**Atg8a-CncC-Dap pathway controls the proteostatic checkpoint.** How is CncC activated by proteostatic stress, and how does CncC execute the proteostatic checkpoint? Our screen included components of the autophagy machinery and cell cycle regulators, and we found that knockdown of Autophagy-related 8a (Atg8a/LC3)[51] or *dacapo* in ISCs (or heterozygosity for the *dacapo* loss-of-function allele *dap^4*)[52], rescued PQ-induced mortality in flies expressing Htt^Q138 (Fig. 3a–c). *Dacapo (dap)* encodes the fly homologue of cyclin-dependent kinase inhibitors of the p21^CIP/p27^KIP family[33,52–54], and Atg8a is a ubiquitin-like protein required for the formation of the isolation membrane during autophagy.

Knockdown of *Atg8a* or *dap* in ISCs, or heterozygosity for *dap^4*, impaired clearance of *mRFP-Htt^Q138* aggregates 2 weeks after induction (Fig. 3d, f), rescued lineage growth (Fig. 3e, g), and rescued *mRFP-Htt^Q138*-induced inhibition of proliferation (Fig. 3h) confirming that both factors are critical for the induction and/or execution of the proteostatic checkpoint in ISCs.

Beyond its role in autophagy, Atg8a/LC3 has also been found to be involved in the non-canonical activation of Nrf2 through an interaction with Keap1[36]. We therefore asked whether this function is required to stimulate CncC activity in response to protein aggregates in ISCs. Interestingly, immunohistochemistry with an antibody specific for Atg8a revealed that Atg8a protein levels increase in ISCs experiencing proteostatic stress (Fig. 4a). We further found that knockdown of Atg8a resulted in loss of the persistent Nrf2 activation (measured by gstD1::lacZ expression) observed in ISCs exhibiting *mRFP-Htt^Q138* + puncta (Fig. 4b).

To assess the role of Dacapo, we performed immunohisto-chemistry with an antibody specific for *Drosophila* Dacapo[53]. ISCs expressing Htt^Q138, or in which *rpn3* was knocked down,

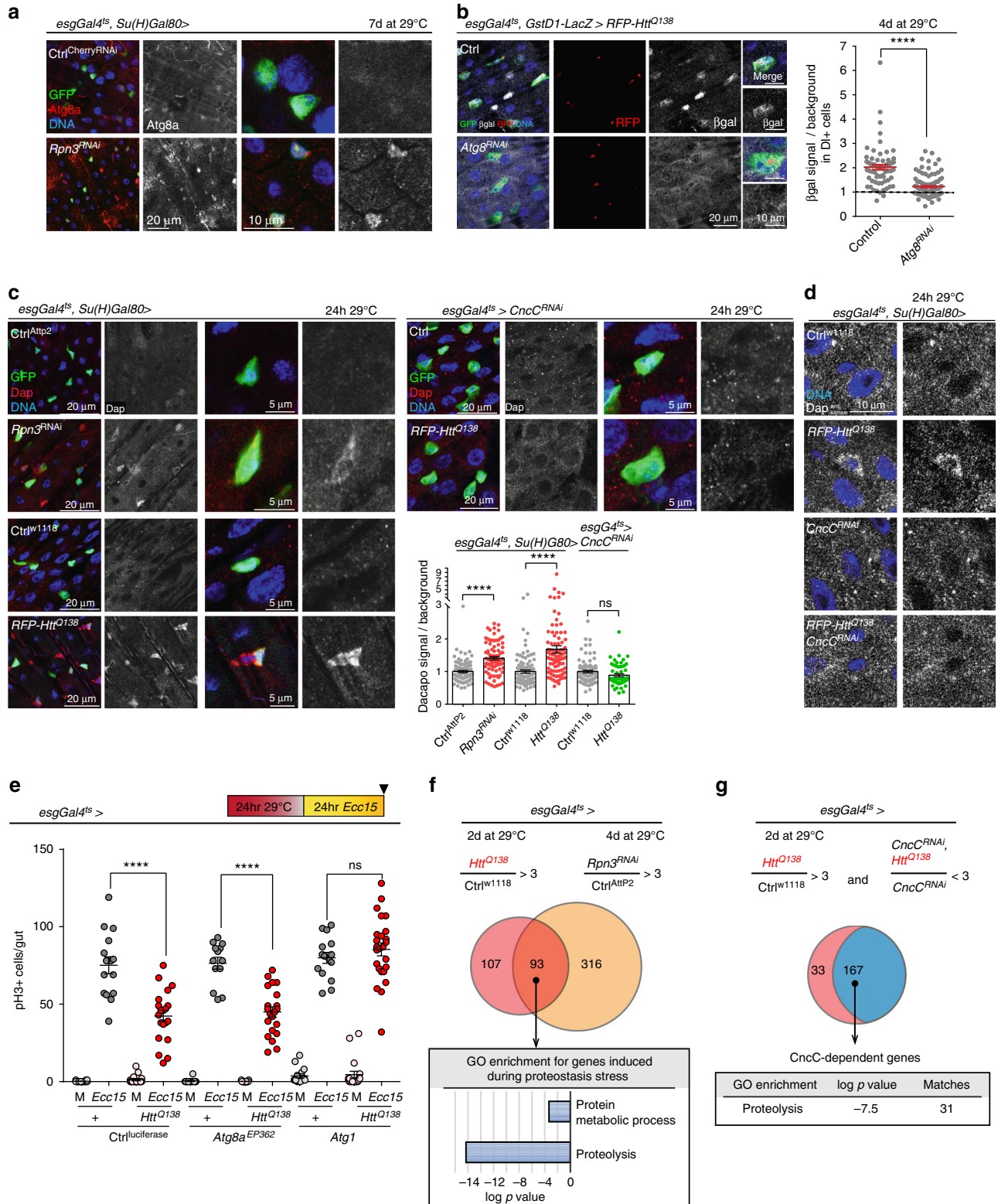

exhibited Dacapo accumulation (Fig. 4c). This accumulation was not observed when CncC was knocked down simultaneously (Fig. 4c). Similar results were observed when fluorescent In situ hybridization was performed in whole guts (Fig. 4d; Supplementary Fig. 2a). These results indicated that Dacapo acts downstream of CncC activation in the proteostatic checkpoint and that CncC regulates *dacapo* at the transcriptional level.

Together with the previously described interaction of Atg8a with Keap1[36], these data suggested that the proteostatic stress-induced increase in Atg8a expression in ISCs promotes CncC activity, triggering the proteostatic checkpoint by CncC-mediated induction of Dacapo. This model predicted that clearance of aggregates by increasing autophagy (e.g., by over-expression of Atg1[55]) should limit the checkpoint, while over-expression of

**Fig. 4** CncC integrates aggregate clearance with cell cycle control to regulate the proteostatic checkpoint. **a** 8-day-old flies expressing Rpn3[RNAi] in ISCs for 7 days at 29 °C. Endogenous Atg8a (red/white), GFP (green), DNA (blue). **b** Flies carrying *gsdtD1::LacZ* reporter and co-expressing *mRFP-Htt*[Q138] and Atg8[RNAi] transgenes for 24 h at 29 °C; βgal (white), RFP (red), GFP (green), and DNA (blue). Graph: βgal signal normalized to background for DI+ cells ($n$ = 67 and 105 from five and eight animals, respectively). Unpaired two-tailed *t*-test. In (**a**) and (**b**), insets show higher magnification of individual ISCs. **c** Flies expressing *Rpn3*[RNAi] or *mRFP-Htt*[Q138] combined with *CncC*[RNAi] in ISCs (green) for 24 h at 29 °C; Dacapo (Dap, red/white). Graph: Dacapo signal in ISCs normalized to background ($n$ = 89, 86, 97, 92, 81, 55 from 5, 5, 8, 9, 4 and 3 animals, respectively, two independent experiments). One-way ANOVA with Sidak's. **d** Fluorescent in situ hybridization of guts from 7-day-old flies after 24 h RFP-Htt[Q138] expression with or without CncC[RNAi]. *dacapo* anti-sense probe in white. Wider gut area and sense probe controls in Supplementary Fig. 2a. **a–d** DNA (blue). Scale bars: 20 μm (**a–c**); 10 μm (insets **a**, **b**; **d**); 5 μm (insets **c**). **e** Expression of *mRFP-Htt*[Q138] combined with *Atg8*[EP362] and *Atg1* in ISCs and EBs of 7-day-old flies for 24 h followed by 24 h of *Ecc15* infection. Graph: pH3 labels mitotic ISCs. One-way ANOVA with Sidak's ($n$ = 16, 15, 19, 20, 12, 13, 15, 21, 15, 15, 19, and 25 guts from independent animals). **f** Venn diagram summarizing the induction of transcripts (at least 3-fold induction and a minimum expression of 10 FPKM) in sorted *esg* + cells (ISCs/EBs; refer to Methods) observed when *mRFP-Htt*[Q138] was expressed for 2 days at 29 °C (red circle, 200 genes total) or *Rpn3*[RNAi] for 4 days at 29 °C (orange circle, 409 genes total). GO analysis for the 93 upregulated genes shared between these conditions. **g** Venn diagram summarizing the overlap between the 200 genes upregulated when *mRFP-Htt*[Q138] was expressed (shown in **c**) and 167 genes that were not induced after *mRFP-Htt*[Q138] expression (<3-fold difference) when *CncC*[RNAi] was co-expressed for 2 days at 29 °C. GO analysis for these 167 CncC-dependent genes is shown. All graphs: Means and s.e.m.; ns not significant, ****$P < 0.0001$

Atg8a should not. We tested this prediction and found indeed that Atg1, but not Atg8a, over-expression prevented the Htt[Q138]-induced inhibition of ISC proliferation after *Ecc15* infection (Fig. 4e).

To explore the transcriptional program downstream of CncC in ISCs in more detail, we performed RNAseq analysis on FACS-purified wild-type or CncC-deficient *escargot*-positive cells (ISCs/EBs) in which the proteostatic checkpoint was engaged through the expression of Htt[Q138] or through the knockdown of *rpn3*. While knockdown of r*pn3* resulted in a more pronounced and broader transcriptional response than the expression of Htt[Q138] (Fig. 4f), 93 of the 200 genes that were induced by Htt[Q138] expression (using a cut-off of three-fold induction and a minimum expression of 10 FPKM) were also induced by knockdown of *rpn3*, illustrating a broad overlap of transcriptional changes elicited by the two conditions (Fig. 4f). Induction of 167 of those 200 genes was dependent on CncC (Fig. 4g). GO analysis of either the 93 genes induced in both stress conditions, or of the 167 CncC-dependent genes revealed a significant enrichment of genes encoding proteins involved in proteolysis and protein metabolism (Fig. 4f, g; Supplementary Fig. 2). These results indicate a role for the proteostatic checkpoint in inducing protein turnover by stimulating the expression of proteases and proteasome subunits in ISCs (Supplementary Fig. 2b, c). Reduction of Rpn3 levels resulted in an upregulation of all proteasome subunits involved in the 20S core, and the 19S base and lid (Supplementary Fig. 2d). Confirming these transcriptome profiling results, we observed a CncC-dependent induction of Rpn3 protein in ISCs and EBs bearing *mRFP-Htt*[Q138] puncta (Supplementary Fig. 2e). Interestingly, the *rpn3* gene contains CncC binding sites according to ChIP-Seq data generated by the ModEncode project[56], suggesting that *rpn3* is a direct CncC target.

**Decline of the proteostatic checkpoint in aging flies**. Aging results in a loss of basal CncC activity in ISCs, limiting pro-liferative homeostasis and causing epithelial dysplasia[26]. We wondered whether this age-related dysfunction is associated with elevated proteostatic stress in ISCs of old animals.

As a measure for endogenous proteostatic capacity in ISCs, we monitored the aggregation of an mCherry-RFP-Rho1 fusion protein expressed under the control of the endogenous Rho1 promoter (herein referred to as mChFP-Rho1)[57], as Rho1 has been found to be highly aggregation prone in aging *C. elegans*[58]. We found that ISCs of aging flies accumulate mChFP + puncta, indicating an increase in aggregated Rho1 (Fig. 5a; intensity of the

mChFP signal across each cell was used as a metric for Rho1 accumulation, as the observed puncta were very bright, yet heterogeneous in size and shape). This aggregation is associated with changes in proteasome activity: GFP[CL1], a GFP fused with a degradation signal to induce its proteosomal degradation[59,60], fails to be degraded in ISCs of aging flies (Fig. 5b). An age-related decline in proteostatic capacity was further supported by the fact that clearance of *mRFP-Htt*[Q138] puncta was impaired in ISCs from old flies (Fig. 5c). Proteostatic responses thus seem to be impaired in ISCs of old flies, and we asked whether this impairment may contribute to the age-related ISC hyperpro-liferation observed in the *Drosophila* intestine. Indeed, lineage-tracing and *Ecc15* infection experiments showed that ISCs in old flies proliferate even in the presence of *mRFP-Htt*[Q138] (Fig. 5d, e; Supplementary Fig. 3). Consistently, the induction of *dacapo* expression by Htt[Q138] was impaired in old ISCs (Fig. 5f).

Since these results suggested that the CncC-mediated proteo-static checkpoint is deficient in ISCs of old flies, we asked whether its activity could be restored by the drug Oltipraz, an activator of Nrf2/CncC[29]. Strikingly, Oltipraz exposure was sufficient to promote proteasome activity in ISCs of old flies (as measured by the degradation of GFP[CL1]; Fig. 6a, Supplementary Fig. 4), and to reduce the accumulation of endogenously aggregating proteins like Rho1 (Fig. 6b). CncC or Atg8 over-expression specifically in ISCs also reduced the accumulation of poly-ubiquitinated protein aggregates in ISCs (Fig. 6c), and old flies that were exposed to Oltipraz after the expression of Htt[Q138] cleared *mRFP-Htt*[Q138] puncta more efficiently than control animals (Fig. 6d). This was accompanied by a restoration of the inhibition of ISC prolifera-tion after proteostatic stress (Fig. 6e).

Over-expression of CncC promotes epithelial homeostasis in old flies[26], and we find that over-expression of CncC or Atg8a in ISCs also limits the age-related increase in barrier dysfunction: old animals (50 days old) in which CncC or Atg8a were over-expressed for an extended period of time in ISCs, showed a significant reduction in the leakage of a non-absorbable blue food dye into the blood-like hemolymph ('Smurf' assay)[61,62] (Fig. 6f). We asked whether the activation of the proteostatic checkpoint would also extend lifespan and found that intermittent, late-life Oltipraz treatment was sufficient to significantly extend lifespan of wild-type animals, and of animals that had been exposed to proteostatic stress in ISCs by transient over-expression of Htt[Q138] (Fig. 6g). While this suggested that activating the proteostatic checkpoint is limiting for longevity, these experiments could not differentiate between lifespan extension driven by local effects of Oltipraz on gut homeostasis, or driven by systemic effects of Oltipraz. However, we found that Oltipraz treatment also

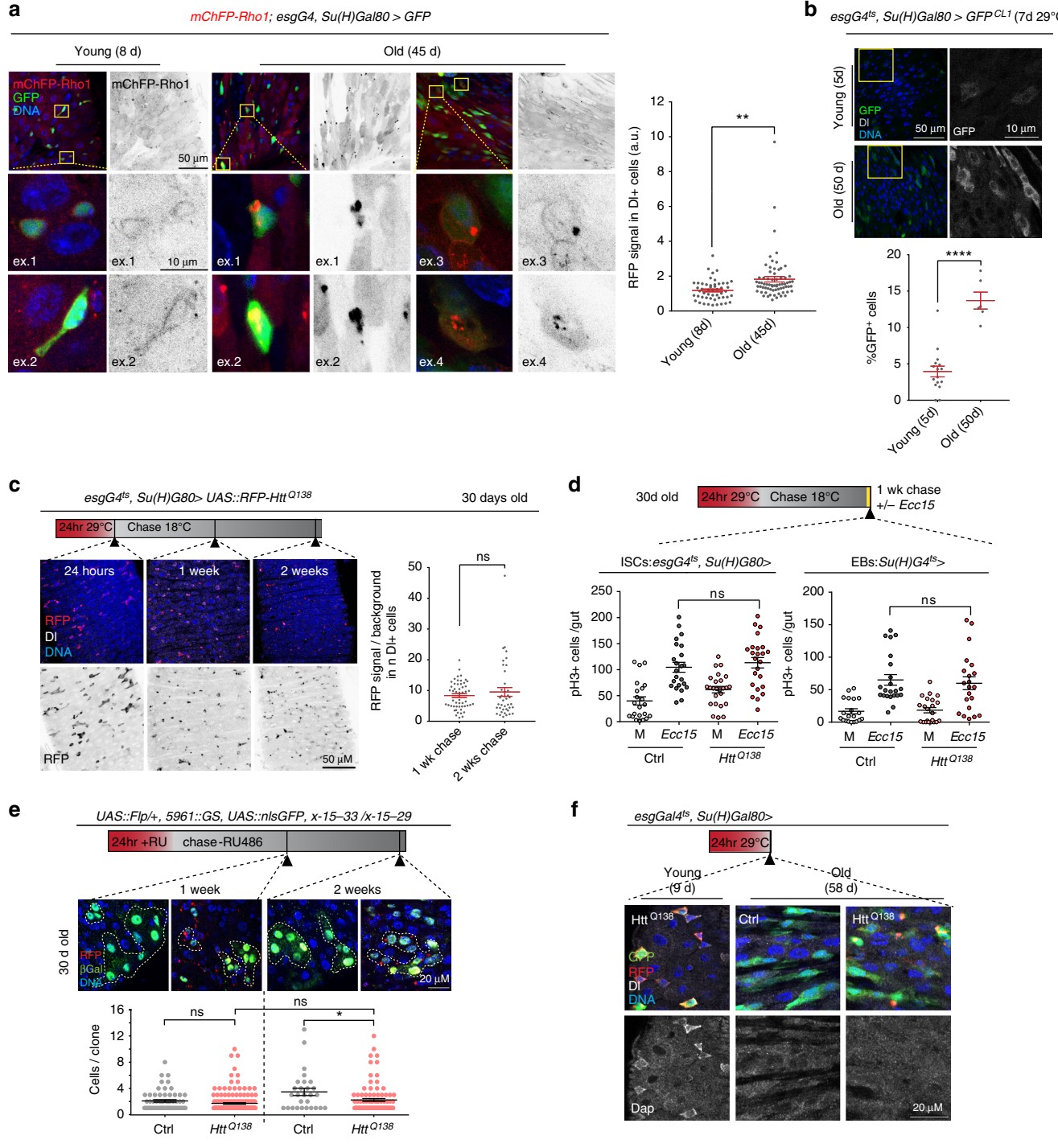

significantly slowed the progression of intestinal epithelial barrier dysfunction in aging flies (Fig. 6h). These results support the idea that the age-related loss of the proteostatic checkpoint in ISCs contributes to increased mortality in aging flies. Accordingly, stimulating proteostatic capacity in SCs is beneficial for tissue health and longevity (Fig. 7).

## Discussion

Our work identifies and characterizes a proteostatic checkpoint that allows adjusting SC activity to proteostatic challenges in the intestinal epithelium. We posit that the coordinated control of aggregate clearance and cell cycle progression is critical to ensure long-term tissue homeostasis, prevent the accumulation of damaging protein aggregates in ISCs, and protect differentiated cells from inheriting such aggregates. This process declines with age, likely contributing to the age-related loss of tissue homeostasis.

The central role of Nrf2/CncC in the proteostatic checkpoint is consistent with its previously described and evolutionarily conserved influence on longevity and tissue homeostasis[26,29], and is likely to be conserved in mammalian SC populations, as Nrf2 has for example been shown to influence proliferative activity, self-renewal and differentiation in tracheal basal cells[27]. It may be

**Fig. 5** ISCs from old flies fail to activate the proteostatic checkpoint. **a** Young (8 days) or old (45 days) flies expressing mCherryRFP-Rho1 (red/black) under the control of its endogenous promoter; GFP (green). Detailed panels ('ex.') showing examples of individual ISCs. Graph: RFP signal normalized to background ($n = 51$ and 69 Dl+ cells from five and eight animals, respectively from two independent experiments). Two-tailed unpaired $t$-test. **b** Flies reared at 18 °C were placed for 7 days at 29 °C when they were 5-days old (young) or 50-days old (old) to express the proteasome reporter GFP$^{CL1}$ specifically in ISCs. % GFP-positive cells within an area of the posterior midgut are shown ($n = 16$ and 6 guts from independent animals). Two-tailed unpaired $t$-test. **c** Experiment as in Fig. 1c, but with 30-day-old flies. RFP signal in Dl+ cells normalized to background ($n = 52$ and 40 from three and three animals respectively, one representative of two independent experiments). mRFP-Htt$^{Q138}$ aggregates in red/black. Two-tailed unpaired $t$-test. **d** Quantification of the number of mitotic cells (pH3+) per guts of 30-day-old flies in which mRFP-Htt$^{Q138}$ was expressed in ISCs or in EBs for 24 h at 29 °C, placed at 18 °C for 1 week followed by an 8 h Ecc15 infection. pH3+cells per gut ($n = 22$, 23, 25, 24 (first graph); 20, 22, 22, and 21 (second graph) from independent animals). Two-way ANOVA with Tukey's. **e** Experiment as in Fig. 1d, but with 30-day-old flies. ISC lineages labelled with anti-βGal (green, yellow outlines), carrying or not mRFP-Htt$^{Q138}$ aggregates (red). One-way ANOVA with Sidak's ($n = 61$, 176, 28, 96 from 5, 10, 4, and 10 independent animals respectively). **f** Young (9 days) or old (58 days) flies expressing or not mRFP-Htt$^{Q138}$ transgene in ISCs for 24 h at 29 °C. Dacapo (white), RFP (red). **a**, **b**, **c**, **e**, **f** Images show representative areas in posterior midgut, DNA (Hoechst, blue), experimental timeline indicated. Scale bars: 50 μm (**a**–**c**); 10 μm (insets **a**, **b**); 20 μm (**e**, **f**). Means and s.e.m. shown in all graphs; ns not significant, ****$P < 0.0001$, **$P < 0.01$, *$P < 0.05$. Genotypes in Supplementary Table 1

unique to somatic SCs, however, as CncC or Nrf2-mediated inhibition of cell proliferation is not observed during development (such as in imaginal discs[26]) or in other dividing cells[63]. Assessing the existence of an Nrf2-induced proteostatic checkpoint in mammalian SC populations will be an important future endeavor.

Mechanistically, our results support a model in which the presence of protein aggregates activates CncC through Atg8a-mediated sequestration of Keap1[36]. In mammals, Nrf2 activation can also be achieved through the interaction of Keap1 with the Atg8a homologue LC3 and p62[32,36], and ref [2]p/p62 contributes to the degradation of polyQ aggregates in Drosophila[64], suggesting that a conserved Atg8a/p62/Keap1 interaction may be involved in the activation of the proteostatic checkpoint. The activation of CncC after cytosolic proteostatic stress described here thus differs mechanistically and in its consequence from the regulation of CncC after other types of protein stress in ISCs: in response to unfolded protein stress in the ER, CncC is specifically inactivated by a ROS/JNK-mediated signaling pathway[24,26]. This mechanism allows ISC proliferation to be increased in response to tissue damage, but can also contribute to the loss of tissue homeostasis in aging conditions[25]. The activation of CncC after cytosolic protein stress, in turn, allows arresting ISC proliferation during protein aggregate clearance. The distinct responses of ISCs to cytosolic or ER-localized proteostatic stress has interesting implications for our understanding of the maintenance of tissue homeostasis. While the XBP1-mediated UPR-ER allows the expansion of the ER and the induction of ER chaperones to deal with a high load of unfolded proteins in the ER, it also stimulates ISC proliferation through oxidative stress and the activation of PERK and JNK. It is tempting to speculate that the sequestration of unfolded proteins within the ER allows ISCs to proceed through mitosis without the possibility of major misregulation, while the presence of cytosolic protein aggregates may be a unique danger to the viability of the cell and its daughters. It seems likely that constitutive activation of autophagy and proteasome pathways during the clearance of cytosolic aggregates is incompatible with the need for intricate regulation of these same pathways during the cell cycle in proliferating ISCs. It will be of interest to explore this hypothesis further in the future.

Our data suggest that the coordination of cell cycle arrest and aggregate clearance is achieved by the simultaneous induction of the cell cycle inhibitor Dacapo and a battery of genes encoding proteins involved in proteolysis. While we were able to detect *dacapo* transcript expression in ISCs by fluorescent in situ hybridization at 24 h after Htt$^{Q138}$ expression, it remains unclear whether Dacapo is induced directly by CncC or via the action of a CncC target gene. It is surprising that we have not observed

transcriptional induction of autophagy genes in our RNAseq experiment, but it is possible that this is due to the fact that we only sampled one timepoint after induction of protein aggregates. Since the transcriptional response of autophagy genes is likely very dynamic, a more time-resolved transcriptome analysis during aggregate formation and clearance may have captured such a response.

It is further notable that *dap* deficient ISC clones exhibit a significantly higher aggregate load in our experiments than wild-type ISC clones. This suggests that the induction of proteolytic genes and of cell cycle regulators is not only coincidentally linked by CncC, but that aggregate clearance and the cell cycle arrest mediated by Dacapo need to be tightly coordinated for effective ISC proteostasis. It will be interesting to explore the mechanism of this requirement in the future. It is tempting to speculate that, as the elimination of protein aggregates requires an increase in proteasome activity, and proteasome activity can influence cell cycle timing[65], cell cycle inhibition is a critical safeguard against de-regulation of normal cell cycle progression.

Our data suggest that Atg8a induction in ISCs experiencing proteostatic stress may serve a dual purpose—sustained activation of the proteostatic checkpoint as well as increased autophagy flux. This dual role is distinct from other autophagy components like Atg1, since Atg1 over-expression, an efficient way of promoting autophagy in *Drosophila* cells[55], counteracts the checkpoint rather than promoting it. Exploring the relative kinetics of Atg8a and Atg1 induction in ISCs after proteostatic stress is likely to provide deeper mechanistic insight into the regulation of the proteostatic checkpoint.

Critically, the proteostatic checkpoint is reversible. Based on our data and previous studies, we propose that upon clearance of aggregates, the Keap1/Atg8a interaction is decreased, thus releasing Keap1 to inhibit CncC[36] (Fig. 7a). Our lineage-tracing studies show that this allows re-activation of ISC proliferation and recovery of normal regenerative responses.

The loss of proteostatic checkpoint efficiency in ISCs of old guts is likely a consequence of the age-related inactivation of CncC in these cells (possibly caused by chronic oxidative stress[26,29]) (Fig. 7b). Accordingly, reactivating Nrf2/CncC in the gut by overexpressing CncC is sufficient to restore epithelial homeostasis in the intestine of old flies[26], and we find that exposing animals to Otipraz intermittently late in life promotes epithelial barrier function and extends lifespan.

Since Nrf2/CncC and other components required for the proteostatic checkpoint are conserved across species, we anticipate that our findings will be relevant to homeostatic preservation of adult SCs in vertebrates. Supporting this view, mammalian Cdkn1a (p21) has been described as an Nrf2 target gene[66].

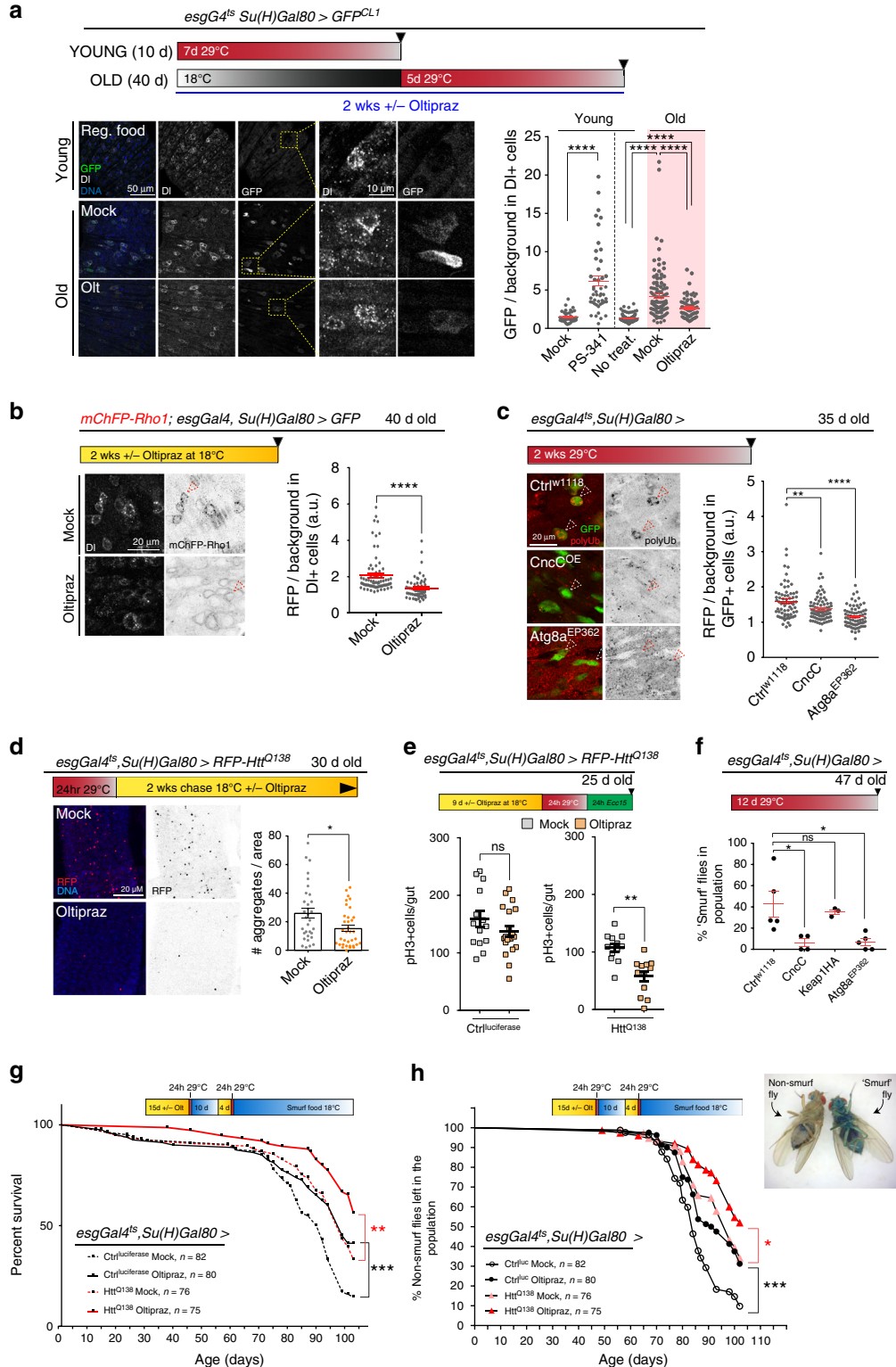

Transient activation of Nrf2 may thus be a viable intervention strategy to improve proteostasis and maintain regenerative capacity in high-turnover tissues of aging individuals.

## Methods

***Drosophila* stocks, culture, and treatments**. Flies were raised on standard yeast/molasses food containing 13.8 g agar, 22 g molasses, 75 g malt extract, 18 g dry inactivated yeast, 80 g corn flour, 10 g soy flour, 6.25 ml, propionic acid, 2 g methyl 4-hydroxybenzoate, 7.2 ml ethanol (200 proof) and water up to 1 L. Flies were

maintained at 25 °C and 65% humidity on a 12 h light/dark cycle, unless otherwise indicated. Only mated (for at least 3 days) female animals were used in all experiments.

The following strains were obtained from the Bloomington Drosophila Stock Center: $w^{1118}$, Ctrl$^{Attp2}$ (P{CaryP}attP2, BL36303), Rpn3$^{RNAi}$ (P{TRiP.HM05247}attP2, BL30503), Rpn2$^{RNAi}$ (P{TRiP.HMS00533}attP2, BL34961), RPT6r$^{RNAi}$ (P{TRiP.HMS01330}attP2, BL34342), dap$^4$ [52] (BL6639), dap$^{RNAi}$ (P{TRiP.HMS01610}attP2, BL36720)[67], UAS::Atg1 (BL60734)[68], mChFP-Rho1 (P{mChFP-Rho1}10, w*, BL52280) and Atg8$^{EP362}$ (P{EP}Atg8aEP362, BL10107). The following lines were obtained from the Vienna Drosophila RNAi Center: CncC$^{RNAi}$ (VDRC 108127, Transformant ID KK101639); Atg8a$^{RNAi}$ (VDRC 109654,

**Fig. 6** The proteostatic checkpoint can be reactivated in old ISCs. **a** Young or old flies of indicated ages and genotypes were treated as indicated. In one group, food was supplemented for 24 h with 40 µM PS-341 or vehicle. Graph: GFP signal normalized to background ($n = 37, 47, 98, 116, 86$ DI+ cells from the guts of 5, 4, 11, 11, and 11 animals, respectively, from two independent experiments). Young mock vs. PS-341: Two-tailed unpaired $t$-test. Young and old +/− Oltipraz: one-way ANOVA/Sidak's. **b** Old flies expressing mCherryRFP-Rho1 (black) under the control of its endogenous promoter, treated as indicated. Graph: RFP signal normalized to background ($n = 75$ and 65 DI+ cells from the guts of four and five animals, from two independent experiments). Two-tailed unpaired $t$-test. **c** Polyubiquitin (red/black) in posterior midgut of flies of indicated genotypes and treatments. RFP signal normalized to background ($n = 72, 84$, and 71 GFP+ cells from the guts of 8, 7 and 7 independent animals, respectively). One-way ANOVA/Sidak's. **d** RFP + aggregates (red/black) in posterior midguts of flies with indicated genotypes and treatments. Graph: Number of aggregates per area ($n = 32$ and 35 animals from three independent experiments). Unpaired two-tailed $t$-test. **a–d** Images show representative areas in posterior midgut, DNA (Hoechst, blue). Scale bars: 50 µm (**a**); 10 µm (insets **a**); 20 µm (**b–d**). **e**, pH3+ cells/gut in flies treated as indicated ($n = 14, 19, 12$, and 12 guts of independent animals, respectively). Unpaired two-tailed $t$-test. **f** Percentage of 'Smurf' flies aged for 35d at 18 °C, then 12d at 29 °C on regular food, before being placed on 'Smurf' food for 6d at 29 °C ($n = 5, 4, 3, 5$, replicates from two experiments). One-way ANOVA. Example of 'non-smurf' and 'smurf' flies shown. **a–f** Means and s.e.m. shown in all graphs. **g** Survival curve of flies reared and treated at 18 °C with 500 µM Oltipraz or vehicle for the indicated time (see timeline and refer to Methods). **h** Percentage of non-smurf flies treated as in 'g' (refer to Methods). **g**, **h** Number of flies used indicated in figure; Mantel-Cox log-rank test. For all graphs, ns not significant, ****$P < 0.0001$, ***$P < 0.001$, **$P < 0.01$, *$P < 0.05$

Transformant ID KK102155) tested previously in[69]; and attP[Ctrl] (KK lines RNAi control, VDRC 60100). The following lines were gifts: UAS::mRFP-Htt[Q138] [43]) from T. Littleton (MIT, Boston MA), GstD1::LacZ, CncC[VL110] and UAS::Keap1HA[29] from D. Bohmann (University of Rochester, USA); X-15-29 and X-15-33[47] from N. Perrimon (Harvard, USA); 5966::GeneSwitch[70] from B. Ohlstein (Columbia University, USA), esg::Gal4,Su(H)GBE::G80,UAS::2XEYFP,tub::G80[ts 46] from S. Hou (NIH, USA), UAS::GFP-CL1 from Udai Pandey (University of Pittsburg).

See Supplementary Table 1 for specific genotypes used in each experiment.

For the conditional expression of UAS-linked transgenes we used the TARGET system[46]. The esg::Gal4 driver was combined with Gal80, a temperature-sensitive Gal4 inhibitor driven by a tubulin promoter (i.e., tub::Gal80[ts]). All fly crosses were maintained at 18 °C to avoid gene expression. Progeny of these crosses were collected and aged at 18 °C. Mated female's progeny with the indicated age were shifted to the restrictive temperature of 29 °C for the time indicated to allow gene expression. For chase experiments, flies were placed at 29 °C for 24 h then shifted back to 18 °C and dissected at various time points.

For RU486 (mifepristone) food supplementation, 100 µl of a 5 mg/ml solution of RU486 in ethanol 80% or vehicle (ethanol 80%) was deposited on top of the food and dried for at least 16 h to ensure complete evaporation, resulting in a 0.2 mg/ml concentration of RU486 in the food accessible to flies (previously tested in ref. [22]). Flies kept at 25 °C were fed on RU486 food or mock for 24 h and dissected at 1 week or 2 weeks after treatment.

For Oltipraz (LKT laboratories, Inc. Cat. No. O4578) food supplementation, 10 ml solution of 50 mM Oltipraz diluted in 10% DMSO diluted in water (or for mock treated, 10% DMSO in water) was added to 1 L of melted fly food (cooled to ~65 °C) for a final concentration of 500 µM Oltipraz in 0.1% DMSO. Food was mixed well and poured into vials. Flies were aged at 18 °C on regular food, and at the indicated age flies were fed food with or without Oltipraz for the time indicated at 18 °C. Food was replaced every 2–3 days.

PS-341 (also known as Bortezomib, APExBIO Cat. No. A2614) was used in young flies as positive control since it inhibits proteasome activity. For PS-341 treatment the protocol from[39] was slightly modified. Briefly, 100 ml of fly food was melted and divided in two flasks: to 48 ml of melted food (cooled to ~65 °C) it was added 2 ml of a 1 mM solution of PS-341 diluted in water (to obtain a final concentration in food of 40 µM PS-341) and to the other 48 ml it was added 2 ml of water only (mock food). Food was mixed well and poured into vials. Flies were aged at 18 °C until the age indicated and placed for 6 d at 29 °C on regular food, and then treated with food +/−PS-341 for 24 h at 25 °C.

For the 'Smurf' assay, the protocol from[61,62] was slightly modified. Briefly, regular fly food was melted and mixed with FD&C blue dye #1 for a final concentration of 2.5% (w/v). Flies at the desired aged were switched from regular food to 'Smurf' food for 6 consecutive days, counting the number of Smurf flies per vial every day. Food was replaced every 2 days.

Lifespan assay was done following protocol from ref. [71]. Briefly, to calculate survival curves and percentage of non-smurf flies in a population, flies were reared at similar densities at 18 °C, mated for three days, and then females were separated into vials with regular food while aging at 18 °C. When flies were 30 days old, these were switched to food containing 500 µM Oltipraz or vehicle (recipe explained above) for 15 days. After treatment, flies were put back on regular food and placed at 29 °C for 24 h to induce transgene expression, and then placed back on regular food containing blue dye for 10 days at 18 °C. This protocol was repeated at 56 days of age (4 days Oltipraz and 24 h 29 °C). Dead flies were counted every three days, and the number of live 'smurf' flies were quantified accordingly. To obtain the percentage of non-smurf flies in populations of flies treated with Oltipraz or vehicle, quantification of smurf flies started right after feeding Smurf food for the first time (from 46 to 102 days old).

This study complies with all relevant ethical regulations for animal testing and research required for the use of *Drosophila melanogaster* as an animal model. As per NIH regulations, no ethical approval was needed for work with *Drosophila melanogaster*.

**Immunohistochemistry and microscopy**. Intact guts from adult female flies were dissected in PBS, fixed in fixative solution (4% formaldehyde in a pH 7.5 solution containing 100 mM glutamic acid, 25 mM KCl, 20 mM MgSO₄, 4 mM sodium phosphate dibasic, 1 mM MgCl₂), washed twice in wash buffer (1× PBS, 0.5% bovine serum albumin and 0.1% Triton X-100) first for 10 min at room temperature (RT), and then for 1 h at 4 °C in a shaker. Primary and secondary antibodies were incubated overnight at 4 °C or for 4 h at RT. All antibodies were diluted in wash buffer. For the immunostaining of Delta antibody we used a methanol-heptane fixation method described in ref. [72] followed by the above described wash steps and antibodies incubations. Hoechst was used to stain DNA. All Images were taken on a Zeiss LSM 710 or a Leica SP5 confocal microscope and processed using Adobe Photoshop, Illustrator and FIJI is just Image J (FIJI).

**Antibodies**. Commercial primary antibodies and dilutions were used as following: mouse anti-mono- and polyubiquitinylated proteins, 1:100 (FK2 clone, Enzo Cat. No. BML-PW8810); rabbit antiphospho-Histone H3 Ser10, 1:1000 (EMD Millipore Cat. No. 06-570); mouse anti-Rpn3/PSMD3 (G-1), 1:500 (Santa Cruz Biotecnology Inc Cat. No. sc-393588); Rabbit anti-β-galactosidase (Cappel MP Biomedicals, 1:500); mouse anti-Armadillo (DSHB, 1:100), mouse anti-Prospero (DSHB, 1:100) and mouse anti-Dacapo (DSHB, 1:4). Rat anti-Delta antibody, 1:500, was a gift from M. Rand (University of Rochester, USA). Rabbit anti-Atg8a, 1:200, was a gift from G. Juhasz (Eotvos Lorand University, Hungary). Fluorescent secondary antibodies were from Jackson Immunoresearch.

**Fluorescence quantification**. Fluorescent quantification was performed in FIJI ImageJ by selecting an ROI, measuring the mean intensity of that area, and subtracting background intensity for the respective channel in an area next to the cell quantified. For the quantification of protein aggregates, first an ROI with a fixed area in the fly posterior midgut was selected and the number of aggregates per area was quantified using FIJI cell counter plug in. Each graphed point represents the sum of aggregates from the three individual ROIs per fly posterior midgut.

**Fluorescent in situ hybridization**. Seven wandering L3 larvae (genotype: esg::Gal4[ts]) were collected, homogenized and its RNA extracted using RNeasy Plus Mini Kit Qiagen (Cat. No. 74134). cDNA synthesis was done using iScript cDNA synthesis kit. Primers used to amplify *dacapo* cDNA to generate a probe expanding exon–exon junctions were: Forward 5′-AGCGCCATCAAGAACTGGCCACG-3′ and Reverse 5′-CTCCAGCGGTGGATTTGGGTTGG-3′. The 510 base pair PCR product was cloned into a pCRII-Topo vector using a TOPO TA Cloning kit (Invitrogen Cat. No. K4600-01) and directionality of insert was verified by Sanger sequencing. Vector was linearized with EcoRV or SpeI. (New England Biolabs) Transcription of *dacapo* and fluorescent labelling of antisense and sense probes was done following manufacturer's instruction of the FISH Tag RNA Multicolor kit (Invitrogen Cat. No. MP 32956). In situ hybridization was performed using a protocol adapted from the one suggested in FISH Tag RNA Multicolor kit (Invitrogen Cat. No. MP 32956).

**Cell sorting and RNA-seq analysis**. Intact guts from 7-day-old adult female flies (~100 guts per sample) were dissected in cold PBS supplemented with 1% bovine serum albumin and 2% fetal bovine serum. Tissues (~100 guts per sample) were trypsinized twice using 0.5% Trypsin EDTA (Gibco Cat.No.15400054) for 30 min on rocker at RT each time, and at each step all cells in suspension were collected. All recovered cells were pooled and washed in cold PBS supplemented with 1% bovine serum albumin and 2% fetal bovine serum. Cells in suspension were passed

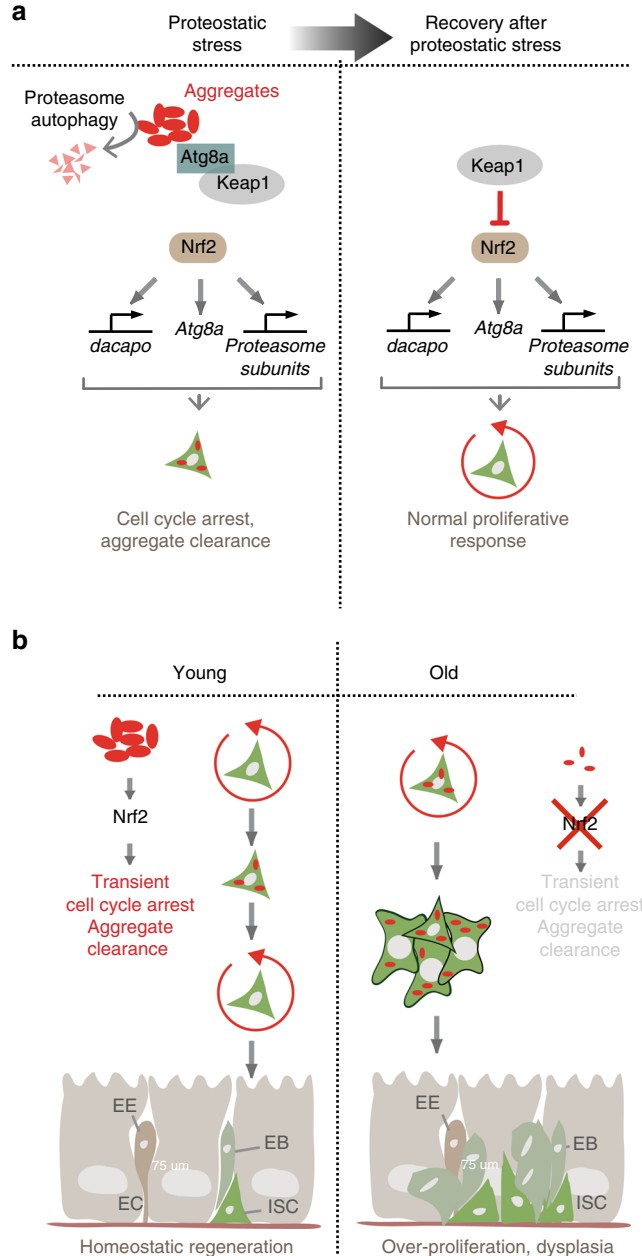

**Fig. 7** A proteostatic checkpoint in intestinal stem cells. **a** Proposed model for activation of the proteostatic checkpoint in ISCs from young flies during proteostatic stress (i.e., presence of protein aggregates). The autophagy protein Atg8a is needed for Nrf2/CncC activation, and is proposed to act by sequestering Keap1. Activated Nrf2/CncC, leads to the upregulation of Atg8a, and dacapo, a cell cycle inhibitor, as well as genes encoding for proteins involved in aggregate clearance such as proteasome subunits. This Nrf2-mediated upregulation leads to a transient cell cycle arrest and aggregate elimination. Once aggregates are cleared, Nrf2 activity is reduced to basal levels and ISCs return to normal proliferative responses. **b** Proposed role of the proteostatic checkpoint in maintaining epithelial homeostasis. Coupling of a transient cell cycle arrest with aggregate clearance allow maintaining epithelial homeostasis under stress conditions. In older animals, this mechanism breaks down due to inactivation of Nrf2/CncC, resulting in dysplasia

through the fine mesh of a cell-strainer cap of a 5 ml polystyrene-round bottom tube (Cat. No.Falcon 352235) and then all GFP-positive cells (GFP was driven by *escargot* promoter expressed in ISCs and EBs) were sorted using FACS. Total RNA was isolated from sorted cells using Trizol reagent, and used as template to generate RNA-seq libraries for Illumina MiSeq sequencing (150 bp paired-end reads were

obtained). Reads were mapped to the *Drosophila* genome (release 5.75) using a standard Tuxedo suite pipeline, and FPKM (Fragments Per Kilobase per Million reads) values were recorded for each gene. Analysis of the data was done in Excel, and Gene Ontology analysis was performed using tools at Flymine.org.

**Paraquat exposure and bacterial infection**. For paraquat exposure experiments, young flies (7–14 days old, raised at 18 °C) were dry fast for 1.5–4 h and then fed 5% sucrose (Mock treated) or 5% sucrose with 5 mM or 7.5 mM Paraquat (Methyl viologen dichloride hydrate, Sigma-Aldrich, Cat. No. 856177) on Whatman filter paper (GE Healthcare Life Sciences Cat. No. 1003-323) above a ¼ piece of Light-Duty Tissue Wipe (VWR Cat. No. 82003-820) for 24 or 20 h, respectively. After treatment, any dead flies were counted, and rest of flies were flipped into a new vial containing normal food and placed at 29 °C for 24 h. After incubation at 29 °C, dead flies were counted. Plotted is the mean (as percentage) of alive flies (i.e., survivors) after 29 °C incubation over total flies treated.

*Ecc15* infection experiments were similarly done as described in ref. [48]. Briefly, *Ecc15* was cultured in LB medium for ~20 h at 30 °C. Flies were dry fast for 2 h and fed 400 µl of concentrated bacteria in 5% sucrose or same volume of 5% sucrose (mock treated) for the time indicated.

**Statistical analyses**. Data was plotted as the mean +/− standard error of the mean. For comparison of only two groups data was analyzed using a student *t*-test and for a comparison of three or more group we performed one-way ANOVA with multiple comparison test. For *Ecc15* infection experiments, and survival experiments we performed a two-way ANOVA with Sidak's or Tukey's multiple comparison test. For the data generated after *Ecc15* infection experiments we performed a Grubbs's Outlier test (also known as Extreme Studentized Seviate method) for each group before analyzing the data using a statistical test. All statistical analyses were performed using Prism (GraphPad Prism v7.02).

**Reporting summary**. Further information on experimental design is available in the Nature Research Reporting Summary linked to this article.

## Data availability
RNAseq data is available at Gene Expression Omnibus (GEO) database under the accession number GSE125385.

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

## Acknowledgements

We thank the members of the Jasper laboratory, especially Pedro Sousa-Victor for helping I.A.R.F. dissect guts for the RNAseq experiment, and Julia McCreary for performing validation experiments. This work was supported by NIH R01s AG047497 and GM117412 to H.J. and NIH T32 AG000266 and F32 AG052275 to I.A.R.F.

## Author contributions

I.A.R.F: designed and performed all experiments except RNAseq experiment. Y.Q. helped dissecting guts for FACS sorting, extracted RNA from sorted cells, prepared libraries and performed RNAseq. H.J. mentored and guided the project. I.A.R.F. and H.J. prepared the figures and wrote the manuscript.

## Additional information

**Competing interests:** The authors declare no competing interests.

