## [Peer Review File · Nature Communications]

Reviewers' comments:

Reviewer #1 (Remarks to the Author):

The paper from Rodriguez-Fernandez, Qi, and Jasper is an interesting, if complex work that provides novel data supporting the existence of a proteostasis checkpoint for *Drosophila* intestinal stem cell (ISC) proliferation. Nearly all the experiments use overexposed Huntington protein (Htt) to generate protein aggregates in the ISCs, a somewhat artificial, but effective treatment. The authors show convincingly that this blocks the ISC cell cycle, and involves CncC (NRF2), its regulator Keap1, and Atg8, and that during ISC arrest the aggregates are cleared. The data also suggests that the checkpoint system weakens with age, and the authors therefore propose that it is causal for age- and gut-dependent mortality, and affects lifespan. This idea is consistent with their previous work on NRF2, but the data presented here don't address the lifespan thesis directly, leaving a big question open. Nevertheless the arrest of the ISC cell cycle by protein aggregates, and involvement of Nrf2 in this process are novel observations in this system that warrant publication. It could quite a bit better if the authors had provided experiments showing that aggregates normally accumulate with age and affect ISC function and lifespan (without Htt induction), or if they had provided an experiment showing that reduction of aggregate formation (by HSPs?) could have beneficial effects. Although I found much of the data and the overall conclusions in the manuscript convincing, there are several instances of weak data, missing obvious experiments, and over-stated conclusions that the authors should attend to in revision to improve the quality of their paper. These are listed below in order of appearance.

1. In Fig 1c & 2d line we see Htt expression and clearance. The authors state that they've scored "aggregates" here and later in the paper, but in fact they've just scored total RFP-Htt signal, which shows up as both puncta and diffuse cytoplasmic staining in their pictures. The text should be amended to accurately reflect that they have not scored only protein aggregates. Line 125, 176.
2. For Fig 1d, please note or show data on clone frequency as well as cell#/clone. If the ISCs are being killed by Htt then there will be fewer clones (as well as fewer cells/clone).
3. Line 137: The text states that the effect is unique to "somatic stem cells" but actually the difference noted is between wing progenitors and ISCs, and generalizing it as a stem cell phenomenon is an overstatement. I suggest removing this conclusion.
4. Line 166. The authors say they did a "genetic screen" for factors involved in the Htt checkpoint, but they don't mention what genes were screened. It would be reasonable to include a list of screened genes and effects in a supplement. Otherwise it's not helpful to mention a screen here.
5. The data in Fig 2b doesn't support the authors' conclusion that loss of CncC released the checkpoint. In fact what has changed here is the effect of Ecc15 in the controls without Htt. The effect of CncC-RNAi in the presence of Htt looks insignificant. Please replace this key data with something that supports the conclusion. Showing an experiment at different ages as in 1e might be helpful.
6. Fig 1e is important but the panels are too small, it's unclear why there is RFP in the 8 panels to the left, and the data should be quantified to better support the conclusion.
7. In Fig 3, the authors should provide data on Dacapo-RNA and/or dap/dap mutant clones, as well as heterozygous mutants. Presenting the dap/+ condition is not very convincing, as it's not clear why this gene should be haplo-insufficient for function, and chromosomal background effects are not ruled out. Overall the conclusion that dacapo induction is the cause of the ISC arrest after Htt expression is not sufficiently supported.

8. Likewise, it's not clear why the authors chose to present Atg8-RNAi data in figure 3c, and atg8/+ data in Fig 3d. Ideally, both types of data would be shown for both genotypes. Otherwise the reader might suspect un-ethical data-selection.

9. The data on Dacapo expression (Fig 4B) is quite messy and only half-way convincing. Would it be possible for the authors to show dap capo mRNA expression or reporter gene data? This would be logical as their model invokes transcriptional control of Dap y Nrf2. In fact, they should have seen dap mRNA induction in their RNAseq data, (Fig 4CD), but it's not mentioned. Why not? Please comment.

10. The Oltopraz experiment in Fig 6 is very interesting, but it's not carried through to logical completion. Did this Nrf2 activator restore the ability of old ISCs to induce Dacapo and suppress ISC proliferation after Htt induction? Did this treatment affect lifespan? Please discuss.

11. The authors propose that Atg8 regulates Nrf2 through a non-autophagy dependent function. If this is true, then other Atg genes should not have similar effects as Atg8. Did the authors test this in their screen, and is this the case? Please discuss.

12. The authors show abundant data indicating that the "proteostasis checkpoint" they characterize is deleterious in the context of damage induced gut epithelial regeneration, but they actually never show direct benefit of this checkpoint to the animal. Although they propose that it may stave off age-dependent dysplasia and extend lifespan (indeed this conclusion is alluded to in the title and abstract) hardly any data directly supporting this is presented here (see only Fig 6c). This important conclusion seems to rest on mainly on their previous work on from their lab on CncC, Keap1, and Jafrac (Hochmuth, 2011, Biteau 2010). In addition, they don't present any data showing that the Htt aggregates they produce are deleterious to gut function or organismal fitness or lifespan, though this is a major assumption that underlies the entire study. I feel the paper could be much compelling if it demonstrated that protein aggregates are deleterious (toxic) in some respect, and if it showed that the proteostasis checkpoint confers a fitness benefit.

13. The authors previously published a paper about CncC and the unfolded protein response ER proteostasis pathway in controlling ISC proteostasis and proliferation (Wang et al PLoS Genet 2014). Oddly, this very relevant paper is not cited or discussed here. The authors should discuss how this previous work overlaps with, or is distinct from, the current study. Does Htt overexpression induce the UPRER?

Reviewer #3 (Remarks to the Author):

In this study, the authors demonstrate a presence of proteostatic checkpoint in *Drosophila* intestinal epithelium stem cells, in which an autophagy-related protein Atg8 activates Nrf2-like transcription factor CncC. The manuscript is well written and easy to follow and the data provided in this MS is consistent. Nonetheless, at least one major issue should be considered before the manuscript is acceptable for publication.

Major specific comments:

Since they identify CncC as a key regulator of the proteostatic checkpoint, the authors focus mainly on the induction of proteasomal subunits and the role some of them, as Rpn2, Rpn3 and Rpt6R, play in the process. However, little attention is paid to the role autophagy may play on it. As the other major mechanism of protein clearance of the cells, especially suited for the degradation of large aggregates

and organelles, autophagy is likely to be involved in this proteostatic checkpoint. The fact that the authors identify autophagy-related gene 8 (Atg8) as the inducer of this checkpoint, and that its knockdown impairs their clearance, points to it. Additionally, Nrf2-activation in mammals can be achieved through the interaction with Keap1 of LC3 (an Atg8-homologue) protein, p62. p62 has also been described to participate in the degradation of polyQ aggregates in *Drosophila* (Saitoh, Y., et al. 2015). However, the process is not even mentioned in the Discussion. I would recommend the authors to check if autophagy is upregulated during this proteostatic checkpoint, and to assess if other autophagy-related proteins, like p62, participate on its induction.

Minor specific comments:

Fig. S1a: bottom-right panel (Rpt6R) quality should be improved, as it has too high background. Some aggregates also seem to accumulate in GFP-negative cells.

Page 4: the authors employ mRFP-Htt to test "the proliferative response of ISCs to protein aggregates directly and without impairing proteasome function". However, bibliography is controversial regarding the proteasomal impairment caused by these protein aggregates. The authors should at least tone down the related sentence.

Page 4: the authors state that "RFP-labeled aggregates quickly formed after expression of mRFP-HttQ138, and could be readily observed throughout the ISC cytoplasm after 1 day.". Aggregates are not so clear in Figure 1c, and the same happens in Figures 2d and 3c. The authors should rewrite their sentence, or choose pictures that adjust better to the text.

Figure 1c: pictures should be turned 90°, so the text does not cover the part of the merge image the authors want to focus on.

Page 5: "these observations indicate that ISCs undergo a transient cell cycle arrest in response to protein aggregate formation, that this arrest is sustained until aggregates are cleared". To confirm that it is until aggregates are removed, and not until a secondary effect generated by the aggregates is over, the authors should check if an accelerated clearance (maybe through autophagy induction) results in a faster recovery from the arrest.

Page 6: "Clones derived from ISCs in which CncC was knocked down or in which Keap1 was over-expressed, grew at a higher rate than wild-type clones, and exhibited increased levels of protein aggregates (Fig. 2c)." The image for Keap1 is not representative of this increase of protein aggregates.

Figure 2e: the images selected are not representative. In the bottom left panel, there is a GFP-positive cell under the middle arrow which has β Gal signal. In the lower right panel, there is a GFP-positive cell with no β Gal signal. I recommend the authors to substitute these images with others in which all GFP-positive cells correspond to their message.

Figure 3c/e: In Figure 3c aggregates presence is checked 2 weeks after expression, but in Figure 3e is 3 weeks after expression. Why is that difference? It is not indicated in the text.

Page 7: "LC3/Atg8a has also been found to be involved in the non canonical activation of Nrf2 through an interaction with Keap1. We therefore asked whether this function is required to stimulate CncC activity in response to protein aggregates in ISCs, and found that knockdown of Atg8 indeed resulted in loss of the persistent Nrf2 activation". If the authors want to check if the interaction between Atg8 and Keap1 is required for CncC activation they should do it specifically impairing this interaction, not by knocking down the protein. A mutant form of LC3 not able to interact with Keap1 would be a better approach.

Figure 4b: why does the CncC-RNAi control have a dotted pattern of Dap, which is not observed in non-silenced cells? This is also partially observed in the previous Ctrl-w1118, but not in Ctrl-Attp2.

Page 7: "These results indicate a role for the proteostatic checkpoint in inducing protein turnover by stimulating the expression of proteases and proteasome subunits in ISCs (Fig. S2a-b)." I find intriguing that there are no autophagy genes induced in this situation. Did the authors check it?

Figure 5a: Despite the microscope images are clear, there are a few points in the old flies sample which are much higher than the average. Is the difference between young and old flies still significant

if the three higher dots of adult flies are not considered?

Methods: the text should be reviewed to make abbreviations homogeneous (hours appear both as h and hr, days is sometimes abbreviated and sometimes not, milliliters appear both as ml and mL, etc.). Other minor corrections should be that oltripaz appears in capitals in Results, but not in Methods, and the text size in lines 347-351 is different from other parts of the text.

Reviewer #4 (Remarks to the Author):

In this manuscript, Jasper and colleagues reported an interesting phenomenon termed “proteostatic checkpoint”. They showed that intestinal stem cells (ISCs) in *Drosophila* undergo a transient cell cycle arrest upon the induction of protein aggregation. They further demonstrated that the proteostatic checkpoint is regulated by autophagy gene *Atg8a*, Nrf2 homolog *CncC*, and cell cycle inhibitor *Dacapo*. They also found that aggregate-induced cell cycle arrest is lost in old ISCs, while over-expression of *CncC* or *Atg8a* preserved intestinal barrier function in old guts. Overall, the manuscript is well prepared and presented. The finding of “proteostatic checkpoint” and its regulation in ISCs during aging is significant and the study uncovers a novel mechanism underlying the maintenance of tissue homeostasis in ISCs during aging and stress. Although most of the conclusions are well supported, I have a few concerns listed below that need to be addressed before the manuscript is considered for publication.

1. Several methods were used to induce proteostatic stress, such as RNAi against *Rpn3* or *Rpn2*, over-expression of mRFP-Htt[Q138]. It is clear that Htt[Q138]-induced aggregation inhibits cell proliferation (under regenerative pressure), and activates Nrf2 signaling in ISCs. However, to demonstrate that the “proteostatic checkpoint” is not specifically triggered by Htt[Q138], it would be great to show that cell cycle arrest and activation of Nrf2 signaling can also be achieved by silencing proteasome component *Rpn3* or *Rpn2*.

2. In Fig. 3, knockdown of *Atg8a* rescued ISC growth, which is probably due to the release of Keap1 and the reduction of Nrf2 activities. Interestingly, *Atg8a* RNAi also impaired aggregate clearance (Fig. 3c) and a slight reduction of ISC growth (Fig. 3d, “+” group). I wonder if *Atg8a* could have two distinct actions on “proteostatic checkpoint”. On the one hand, *Atg8a* knockdown can block autophagosome formation and impair autophagy, which then directly induces protein aggregation and triggers cell cycle arrest. On the other hand, *Atg8a* RNAi can repress Nrf2 via Keap1 and attenuate Htt[Q138]-induced cell cycle arrest. Authors need to address these possibilities in the discussion section.

To further support the role of *Atg8a* in “proteostatic checkpoint”, it would be great to show if proteostatic stress can induce the expression of *Atg8a*. *Atg8a* reporter (mCherry-*Atg8a*) or *Atg8a* antibody (like GABARAP (E1J4E) antibody from Cell Signaling Technology) can be used.

3. Authors found that the “proteostatic checkpoint” machinery is impaired in old ISCs, while over-expression of *CncC* slows down age-induced intestinal barrier dysfunction. However, it is not clear whether over-expression of *CncC* or Nrf2 activator oltipraz can restore “proteostatic checkpoint” machinery and proteostatic stress-induced cell cycle arrest in old ISCs. Authors need to address this possibility by monitoring the cell proliferation upon *Ecc15* treatment or performing lineage-tracing assay in old ISCs expressing both Htt[Q138] and *CncC* (or oltipraz feeding).

Minor issues:

1. Unlike other figures, Figure 4 used capital letters for each panel, instead of lower-case letters.
2. Line 718, remove “e, f”

3. How many fly guts were used in each RNA-seq analysis? How many biological replicates per condition?

4. Authors should include the list of candidate genes identified from their genetic screen on mortality of PQ-treated and Htt[Q138]-expressing flies, unless these data will be used in a different publication.

5. In some of the experiments, flies were placed at 29 degree (e.g., to induce Htt[Q138]), then back to 18 degree. There is some concern of the heat shock response triggered by high temperature. Could the heat shock response confound the proteostasis analysis and aggregate-related ISC proliferation and cell cycle arrest?

In oltipraz feeding experiment, flies were aged at 18 degree for a long period of time. Because flies age at a slower rate under low temperature. Could these results provide any insights into the regulation of proteostatic checkpoint during normal aging in *Drosophila*?

Response to reviewers NCOMMS-17-26472-T

We would like to thank the reviewers for their positive reception of our work and the constructive critique of the manuscript. All three reviewers considered the work to be of significant interest and the presented data to be convincing, but listed a number of concerns that were summarized by the editor in the following major requests:

- 1) Confirm that the phenomena described in this MS are not specific to Htt[Q138].
- 2) Examine whether Nrf2 activation can restore the proteostatic checkpoint in old ISCs.

We have now addressed these and other remaining concerns of the reviewers by performing additional experiments and by editing our discussion. With respect to the major requests listed above, we have responded as follows:

- 1) Confirm that the phenomena described in this MS are not specific to Htt[Q138].

Our originally submitted results already partially addressed this point. We had shown that:

- Similar to HttQ138 expression, poly-ubiquitin aggregates induced by Rpn3 knockdown cause an ISC cell cycle arrest (Fig. 1a-b, Fig. S1a)
- Age-related accumulation of protein aggregates (using Rho1 as an example) can be observed endogenously (Fig. 5a)
- The age-related decline in proteasome activity (Fig. 5b) can be significantly improved when flies are treated with the Nrf2 activator drug Oltipraz (Fig. 6a)

We have now expanded our analysis of the phenotypic consequences of Rpn3 knockdown, showing that:

- Similar to HttQ138 expression, poly-ubiquitin aggregates induced by Rpn3 knockdown increase Nrf2 activity in ISCs (Fig.2f)
- Accumulation of poly-ubiquitin aggregates induced by Rpn3 knockdown leads to increased expression of Atg8a in intestinal stem cells (Fig.4a)

Together with our previously reported results, the newly obtained findings establish that the observed proteostatic checkpoint is in fact a consequence of protein aggregates generally, and not simply a response to HttQ138. Importantly, we used HttQ138 over-expression to demonstrate that the phenotypes observed in *rpn3* loss of function conditions are not simply a consequence of proteasome dysfunction *per se*, but in fact a result of protein aggregate accumulation. We believe that this combination of proteasome loss-of-function and protein aggregate over-expression data thus represents the most compelling indication that the proteostatic checkpoint is a critical endogenous regulatory mechanism in ISCs.

Our data further support the idea of an age-related dysfunction in this checkpoint and that this dysfunction contributes to the age-related breakdown in ISC proteostasis.

- 2) Examine whether Nrf2 activation can restore the proteostatic checkpoint in old ISCs.

Our originally submitted results partially addressed this question by showing that

- Oltipraz treatment can increase proteasomal activity in ISCs of old animals (Fig. 6a).
- Oltipraz treatment can improve the clearance of protein aggregates in ISCs of old animals (Fig. 6c).
- CncC and Atg8a over-expression in ISCs are sufficient to limit barrier dysfunction in old animals (Fig. 6e).

We have now performed a series of experiments to build on these observations and show now that:

- Oltipraz treatment can decrease the age-related accumulation of endogenous Rho1 aggregates in intestinal stem cells (Fig 6b),
- Oltipraz can decrease ISC over-proliferation in old flies (Fig. 6e),
- Overexpression of CncC and Atg8a in ISCs reduces the age-related accumulation of endogenous poly-ubiquitinated aggregates (Fig. 6c).
- We have also tested whether Oltipraz treatment can reduce the prevalence of barrier dysfunction in old flies, and have observed a trend that supports such a conclusion, but our experiments did not reach statistical significance due to technical reasons, and we have thus only included them here for the reviewers: (Fig. R1). Since over-expression of CncC and Atg8a was targeted specifically to stem cells, we believe that these

experiments more convincingly demonstrate the beneficial effect of CncC activation in ISCs than the systemic administration of Oltipraz.

In response to other reviewers' concerns, we have also obtained additional new data points that support our conclusions:

- We find that Dacapo knockdown (by RNAi) impairs aggregate clearance capacity of intestinal stem cells from young animals (Fig.3e)
- We show that Atg1 over-expression is sufficient to reduce the cell cycle arrest in ISCs expressing protein aggregates, indicating that induction of autophagy is sufficient to alleviate proteostatic stress that triggers the proteostatic checkpoint (Fig. 4d). Strikingly, Atg8a over-expression does not reduce the cell cycle arrest, further supporting the notion that Atg8a plays a dual role in the response of ISCs to proteostatic stress: its interaction with Keap1 activates CncC and promotes cell cycle arrest, while its role in the autophagy pathway promotes clearance of aggregates.

In summary, we believe that our new data significantly strengthen the manuscript and provide clear evidence for the role of the newly discovered proteostatic checkpoint in maintaining stem cell function and tissue homeostasis. In the following, we respond in detail to the reviewers' comments:

Reviewer #1 (Remarks to the Author):

The paper from Rodriguez-Fernandez, Qi, and Jasper is an interesting, if complex work that provides novel data supporting the existence of a proteostasis checkpoint for *Drosophila* intestinal stem cell (ISC) proliferation. Nearly all the experiments use overexpressed Huntington protein (Htt) to generate protein aggregates in the ISCs, a somewhat artificial, but effective treatment. The authors show convincingly that this blocks the ISC cell cycle, and involves CncC (NRF2), its regulator Keap1, and Atg8, and that during ISC arrest the aggregates are cleared. The data also suggests that the checkpoint system weakens with age, and the authors therefore propose that it is causal for age- and gut-dependent mortality, and affects lifespan. This idea is consistent with their previous work on NRF2, but the data presented here don't address the lifespan thesis directly, leaving a big question open.

Nevertheless the arrest of the ISC cell cycle by protein aggregates, and involvement of Nrf2 in this process are novel observations in this system that warrant publication. It could quite a bit better if the authors had provided experiments showing that aggregates normally accumulate with age and affect ISC function and lifespan (without Htt induction), or if they had provided an experiment showing that reduction of aggregate formation (by HSPs?) could have beneficial effects.

Although I found much of the data and the overall conclusions in the manuscript convincing, there are several instances of weak data, missing obvious experiments, and over-stated conclusions that the authors should attend to in revision to improve the quality of their paper. These are listed below in order of appearance.

We would like to thank the reviewer for the positive and thoughtful review. We believe that with our new observations, we have addressed the main concern, showing that the proteostatic checkpoint responds to endogenous aggregates, that protein aggregates naturally accumulate with age in ISCs, and that improving proteostasis has beneficial physiological effects:

- We confirm that the phenomena described in this MS are not specific to HttQ138, but that poly-ubiquitin aggregates induced by Rpn3 knockdown induce an ISC cell cycle arrest (Fig. 1a, b), increase Nrf2 activity (Fig. 2f), as well as expression of Atg8a in intestinal stem cells (Fig.4a)
- We show that age-related accumulation of Rho1 aggregates can be observed endogenously (Fig. 5a, these data were already shown in the original manuscript)
- We show that the age-related decline in proteasome activity (Fig. 5b) can be significantly improved when flies are treated with the Nrf2 activator drug Oltipraz (Fig. 6a, these data were already shown in the original manuscript)
- We show that Dacapo knockdown impairs aggregate clearance capacity of intestinal stem cells from young animals (Fig. 3e)
- We show that Oltipraz treatment can (i) decrease the age-related accumulation of endogenous Rho1 aggregates in intestinal stem cells (Fig. 6b), and (ii) decrease the age-related over-proliferation of intestinal stem cells (Fig. 6d).

- We find that overexpression of CncC and Atg8a in ISCs rescues the age-related accumulation of endogenous poly-ubiquitinated aggregates (Fig. 6c).
- We show that over-expression of CncC and Atg8a in ISCs is sufficient to prevent age-related barrier dysfunction (Fig. 6e; these data were already shown in the original manuscript).

Together with previously published work from our lab (Hochmuth et al., 2011, where it was shown that CncC over-expression in ISCs is sufficient to limit age-related epithelial dysplasia), we believe that our data thus conclusively demonstrate that the activation of CncC in response to cytosolic protein aggregates is a critical protective response to preserve tissue homeostasis.

1. In Fig 1c & 2d line we see Htt expression and clearance. The authors state that they've scored "aggregates" here and later in the paper, but in fact they've just scored total RFP-Htt signal, which shows up as both puncta and diffuse cytoplasmic staining in their pictures. The text should be amended to accurately reflect that they have not scored only protein aggregates. Line 125, 176.

We agree with the reviewer, and we have now adjusted the text accordingly. Of note, the relevant scoring was done at time points (1 or 2 weeks after induction) where the RFP-HttQ138 signal is already mostly in aggregates. We therefore chose to measure total RFP signal for accuracy.

2. For Fig 1d, please note or show data on clone frequency as well as cell#/clone. If the ISCs are being killed by Htt then there will be fewer clones (as well as fewer cells/clone).

We have now included these data, showing that clone numbers do not decrease (Fig S1c). Critically, we had already shown that clones can recover from the proteostatic checkpoint, as clones derived from ISCs expressing aggregating proteins were able to regrow to almost wild-type sizes two weeks after the aggregate expression (Fig. 1d; at 2 weeks aggregates have been cleared; Fig. 1c). Since clones are only induced in the first day of the experiment (when the tissue is exposed to RU486), this clearly established that the labeled ISCs were not subject to damage or death, but were able to re-initiate proliferative activity and form normal lineages.

3. Line 137: The text states that the effect is unique to "somatic stem cells" but actually the difference noted is between wing progenitors and ISCs, and generalizing it as a stem cell phenomenon is an overstatement. I suggest removing this conclusion.

We have adjusted this statement.

4. Line 166. The authors say they did a "genetic screen" for factors involved in the Htt checkpoint, but they don't mention what genes were screened. It would be reasonable to include a list of screened genes and effects in a supplement. Otherwise it's not helpful to mention a screen here.

We have in fact screened a number of proteostatic candidates (Atg8, p62, CUL4, DCAF12, HDAC6, and Sirtuins), and we are now listing these in the text.

5. The data in Fig 2b doesn't support the authors' conclusion that loss of CncC released the checkpoint. In fact what has changed here is the effect of Ecc15 in the controls without Htt. The effect of CncC-RNAi in the presence of Htt looks insignificant. Please replace this key data with something that supports the conclusion. Showing an experiment at different ages as in 1e might be helpful.

In these experiments we believe that it is important to compare the effects of HttQ138 expression within the same genetic background (i.e. comparing ISCs with and without HttQ138 in wild-type background, or with and without HttQ138 in CncCRNAi background). We find a significant reduction of proliferation by HttQ138 in the wild-type background, but no effect at all in the CncCRNAi background, supporting the interpretation from the experiment shown in Fig. 2a that CncC activity is critical for the checkpoint.

Since these experiments are very complex (both genetically and in experimental design, requiring precise timing of temperature shifts and infection paradigms), we chose to validate the proposed effect of CncC through the independent experiment shown in Fig 2c, where the effects of CncC inhibition by either RNAi or by Keap1 over-expression on HttQ138-impaired clone growth was tested.

Together with the lack of aggregate clearance in CncC loss of function conditions (Fig. 2d), the activation of CncC by HttQ138 and *Rpn3* knockdown (Fig. 2f), the requirement of CncC in HttQ138-mediated Dacapo induction (Fig. 4c), and the requirement for CncC in the aggregate-induced gene expression program in ISCs (Fig. 4f), we can conclude that CncC is a critical regulator of the proteostatic checkpoint.

6. Fig 1e is important but the panels are too small, it's unclear why there is RFP in the 8 panels to the left, and the data should be quantified to better support the conclusion.

We think the reviewer was referring to Fig 2e. We agree that these images were confusing due to the representation of 4 channels in one panel. We have re-assembled the figures to show GFP/RFP/DNA channels in one panel and the critical channel, showing bGal expression, in a separate channel in gray scale. As requested, we have now also quantified the bGal signal.

We are also including new data showing that *Rpn3* knockdown results in the same activation of CncC signaling as HttQ138 expression (Fig. 2f).

7. In Fig 3, the authors should provide data on *Dacapo*-RNA and/or *dap/dap* mutant clones, as well as heterozygous mutants. Presenting the *dap/+* condition is not very convincing, as it's not clear why this gene should be haplo-insufficient for function, and chromosomal background effects are not ruled out. Overall the conclusion that *dacapo* induction is the cause of the ISC arrest after Htt expression is not sufficiently supported.

We have now performed experiments with *Dacapo* RNAi and show that *Dacapo* knock-down recapitulates the effects of *Dap* heterozygosity (Fig. 3e).

8. Likewise, it's not clear why the authors chose to present *Atg8*-RNAi data in figure 3c, and *atg8/+* data in Fig 3d. Ideally, both types of data would be shown for both genotypes. Otherwise the reader might suspect un-ethical data-selection.

We apologize for this mistake. The figure was mislabeled, as both figures showed data collected with *Atg8a* RNAi. This has now been corrected.

9. The data on *Dacapo* expression (Fig 4B) is quite messy and only half-way convincing. Would it be possible for the authors to show *dap* *capo* mRNA expression or reporter gene data? This would be logical as their model invokes transcriptional control of *Dap* by *Nrf2*. In fact, they should have seen *dap* mRNA induction in their RNAseq data, (Fig 4CD), but it's not mentioned. Why not? Please comment.

We agree that the antibody against *dacapo* is not ideal, but we do believe that it clearly shows up-regulation of *dacapo* protein levels. We have now reanalyzed these data and included much better images. *Dacapo* mRNA is not in fact detected in the RNAseq (its expression is under detection levels in all conditions). To reflect the lack of evidence for transcriptional induction of *dap* by *Nrf2*, we have also modified our cartoon (Fig 7a).

10. The Oltipraz experiment in Fig 6 is very interesting, but it's not carried through to logical completion. Did this *Nrf2* activator restore the ability of old ISCs to induce *Dacapo* and suppress ISC proliferation after Htt induction? Did this treatment affect lifespan? Please discuss.

We have now included additional experiments showing that Oltipraz treatment can (i) decrease the age-related accumulation of endogenous Rho1 aggregates in intestinal stem cells (Fig. 6b), and (ii) decrease the age-related over-proliferation of intestinal stem cells (Fig. 6d).

In the figure included in this rebuttal letter (Fig. R1), we also show evidence that Oltipraz treatment can reduce the incidence of intestinal barrier dysfunction in old flies. Unfortunately, these experiments suffered from unexpected mortality when aging the test populations, resulting in low numbers of individual populations assessed. We therefore didn't reach statistical significance and chose not to include these results in the manuscript.

Lifespan effects by Oltipraz have not so far been reported, and we have not pursued that experiment here, as we believe that Oltipraz will have systemic effects that preclude clear interpretation of the longevity consequences of modulating the proteostatic checkpoint in ISCs. However, it should be noted that Sykiotis and Bohmann have already reported extended lifespan in *Keap1* heterozygous animals.

11. The authors propose that *Atg8* regulates *Nrf2* through a non-autophagy dependent function. If this is true, then other *Atg* genes should not have similar effects as *Atg8*. Did the authors test this in their screen, and is this the case? Please discuss.

This is a good point. The notion that *Atg8a* has a specific role here comes from the findings reported by Jain et al. (2015: J Biol Chem. 290:14945-62), where it is shown that *Atg8a* interacts with p62 and *Keap1* to activate CncC. We

have now included an experiment with Atg1 that indeed shows that the effects of Atg8a and Atg1 on the proteostatic checkpoint are different: Overexpression of Atg1, but not Atg8a, prevents the aggregate-mediated cell cycle arrest (Fig. 4d).

12. The authors show abundant data indicating that the "proteostasis checkpoint" they characterize is deleterious in the context of damage induced gut epithelial regeneration, but they actually never show direct benefit of this checkpoint to the animal. Although they propose that it may stave off age-dependent dysplasia and extend lifespan (indeed this conclusion is alluded to in the title and abstract) hardly any data directly supporting this is presented here (see only Fig 6c). This important conclusion seems to rest on mainly on their previous work on from their lab on CncC, Keap1, and Jafrac (Hochmuth, 2011, Biteau 2010). In addition, they don't present any data showing that the Htt aggregates they produce are deleterious to gut function or organismal fitness or lifespan, though this is a major assumption that underlies the entire study. I feel the paper could be much compelling if it demonstrated that protein aggregates are deleterious (toxic) in some respect, and if it showed that the proteostasis checkpoint confers a fitness benefit.

We now show more extensively that endogenous protein aggregates accumulate in ISCs of aging flies, and that this can be prevented by restoration of CncC activity (Figs. 5 and 6). Together with our demonstration that CncC and Atg8a over-expression prevent barrier dysfunction in old flies (Fig 6e), we believe that these data in fact show that the proteostasis checkpoint is an essential promoter of epithelial homeostasis.

13. The authors previously published a paper about CncC and the unfolded protein response ER proteostasis pathway in controlling ISC proteostasis and proliferation (Wang et al PLoS Genet 2014). Oddly, this very relevant paper is not cited or discussed here. The authors should discuss how this previous work overlaps with, or is distinct from, the current study. Does Htt overexpression induce the UPRER?

We agree, and we have now discussed this paper and the relation of the UPRER with the proteostatic checkpoint in detail. Of note, the UPR ER results in increased oxidative stress, which in ISCs results in a JNK-mediated inactivation of Nrf2, and thus promotes ISC proliferation. The cytoplasmic proteostatic checkpoint described here is thus very distinct from the response reported in Wang et al., 2014.

Reviewer #3 (Remarks to the Author):

In this study, the authors demonstrate a presence of proteostatic checkpoint in *Drosophila* intestinal epithelium stem cells, in which an autophagy-related protein Atg8 activates Nrf2-like transcription factor CncC. The manuscript is well written and easy to follow and the data provided in this MS is consistent. Nonetheless, at least one major issue should be considered before the manuscript is acceptable for publication.

We would like to thank the reviewer for the positive and constructive critique.

Major specific comments:

Since they identify CncC as a key regulator of the proteostatic checkpoint, the authors focus mainly on the induction of proteasomal subunits and the role some of them, as Rpn2, Rpn3 and Rpt6R, play in the process. However, little attention is paid to the role autophagy may play on it. As the other major mechanism of protein clearance of the cells, especially suited for the degradation of large aggregates and organelles, autophagy is likely to be involved in this proteostatic checkpoint. The fact that the authors identify autophagy-related gene 8 (Atg8) as the inducer of this checkpoint, and that its knockdown impairs their clearance, points to it. Additionally, Nrf2-activation in mammals can be achieved through the interaction with Keap1 of LC3 (an Atg8-homologue) protein, p62. p62 has also been described to participate in the degradation of polyQ aggregates in *Drosophila* (Saitoh, Y., et al. 2015). However, the process is not even mentioned in the Discussion. I would recommend the authors to check if autophagy is upregulated during this proteostatic checkpoint, and to assess if other autophagy-related proteins, like p62, participate on its induction.

We fully agree with the reviewer regarding the importance of autophagy in this process, and had therefore already assessed the effects of Atg8 over-expression on the proteostatic checkpoint. We have now complemented these data with an experiment showing that Atg1 over-expression can prevent the proliferation arrest observed in ISCs expressing HttQ138 aggregates (Fig. 4d), suggesting that autophagic clearance of aggregates indeed influences the proteostatic checkpoint. We further now show that Atg8a is induced in aggregate-bearing ISCs (Fig. 4a), further supporting this idea. Interestingly, Atg8a over-expression (as opposed to Atg1 over-expression) does not inhibit the proteostatic checkpoint (Fig. 4d), indicating that, as proposed in the literature (Jain et al. (2015) J Biol Chem.

290:14945-62) and supported by our experiments (Fig. 3a, c, d, 4b), Atg8a induction has a specific role in the response to protein aggregates, activating the proteostatic checkpoint by interacting with Keap1 and stimulating Nrf2 activity.
We have further now more comprehensively discussed the literature regarding autophagy and protein aggregate clearance.

Minor specific comments:

Fig. S1a: bottom-right panel (Rpt6R) quality should be improved, as it has too high background. Some aggregates also seem to accumulate in GFP-negative cells.

We have adjusted image quality, but please note that the right panels are showing Immunostaining against Delta (to mark ISCs). The Delta antibody shows punctuated staining in some cases.

Page 4: the authors employ mRFP-Htt to test “the proliferative response of ISCs to protein aggregates directly and without impairing proteasome function”. However, bibliography is controversial regarding the proteasomal impairment caused by these protein aggregates. The authors should at least tone down the related sentence.

We agree and have now adjusted the text accordingly.

Page 4: the authors state that “RFP-labeled aggregates quickly formed after expression of mRFP-HttQ138, and could be readily observed throughout the ISC cytoplasm after 1 day.”. Aggregates are not so clear in Figure 1c, and the same happens in Figures 2d and 3c. The authors should rewrite their sentence, or choose pictures that adjust better to the text.

We agree and have explained our observations better.

Figure 1c: pictures should be turned 90°, so the text does not cover the part of the merge image the authors want to focus on.

We have now moved the interfering text out of these panels to the top of the image.

Page 5: “these observations indicate that ISCs undergo a transient cell cycle arrest in response to protein aggregate formation, that this arrest is sustained until aggregates are cleared”. To confirm that it is until aggregates are removed, and not until a secondary effect generated by the aggregates is over, the authors should check if an accelerated clearance (maybe through autophagy induction) results in a faster recovery from the arrest.

We have tested this idea and show now that over-expression of Atg1, which in flies induces autophagy (Scott, Juhasz and Neufeld (2007) *Curr Biol.* 17: 1-11), indeed releases ISCs from proliferation arrest (Fig. 4d). This was tested in the context of aggregate-induced inhibition of infection-induced proliferation. It is, however, difficult to exclude a secondary effect induced by the aggregates, as ultimately even such a secondary effect would be reduced by the induction of autophagy. We have explained our reasoning for characterization of the molecular mechanism of the proteostatic checkpoint in this context (page 6).

Page 6: “Clones derived from ISCs in which CncC was knocked down or in which Keap1 was over-expressed, grew at a higher rate than wild-type clones, and exhibited increased levels of protein aggregates (Fig. 2c).” The image for Keap1 is not representative of this increase of protein aggregates.

We have now chosen a different representative image that clearly highlights the high number of aggregates in these clones.

Figure 2e: the images selected are not representative. In the bottom left panel, there is a GFP-positive cell under the middle arrow which has β Gal signal. In the lower right panel, there is a GFP-positive cell with no β Gal signal. I recommend the authors to substitute these images with others in which all GFP-positive cells correspond to their message.

We have followed the suggestion of the reviewer.

Figure 3c/e: In Figure 3c aggregates presence is checked 2 weeks after expression, but in Figure 3e is 3 weeks after expression. Why is that difference? It is not indicated in the text.

In this particular experiment we had included a third timepoint to explore the prevalence of aggregates beyond the two weeks chase, and chose to quantify the latest timepoint. We had also analyzed the data in Fig. 2d at 3 weeks after the induction. We generally don't see a difference between the second and third week, but have more consistent clearance in wild-type conditions after three weeks. We therefore chose to quantify clearance at that timepoint in these experiments. We have explained this technical issue in the Figure Legend.

Page 7: "LC3/Atg8a has also been found to be involved in the non canonical activation of Nrf2 through an interaction with Keap1. We therefore asked whether this function is required to stimulate CncC activity in response to protein aggregates in ISCs, and found that knockdown of Atg8 indeed resulted in loss of the persistent Nrf2 activation". If the authors want to check if the interaction between Atg8 and Keap1 is required for CncC activation they should do it specifically impairing this interaction, not by knocking down the protein. A mutant form of LC3 not able to interact with Keap1 would be a better approach.

We agree that this would be a better approach, but unfortunately such flies are not available. We have now reformulated this passage to better interpret this experiment.

Figure 4b: why does the CncC-RNAi control have a dotted pattern of Dap, which is not observed in non-silenced cells? This is also partially observed in the previous Ctrl-w1118, but not in Ctrl-Attp2.

This is an interesting observation that we had not paid attention to, as we believe that it is due to experimental variation in the staining. We have also reprocessed these data to present clearer representative images.

Page 7: "These results indicate a role for the proteostatic checkpoint in inducing protein turnover by stimulating the expression of proteases and proteasome subunits in ISCs (Fig. S2a-b)." I find intriguing that there are no autophagy genes induced in this situation. Did the authors check it?

We agree that this is surprising, especially given the observation that Atg8a protein is induced in ISCs experiencing proteostatic stress (Fig. 4a). We believe that the transcriptional response to proteostatic stress may be more dynamic than anticipated and that we may therefore not observe changes in autophagy gene expression at the timepoint chosen for the RNAseq. We have now discussed this caveat in the text.

Figure 5a: Despite the microscope images are clear, there are a few points in the old flies sample which are much higher than the average. Is the difference between young and old flies still significant if the three higher dots of adult flies are not considered?

We agree that this is a concern, and we have performed a Grubb's Outlier test to assess this question. For the "young" sample the test indicated that the value 3.17 is a significant outlier ($P < 0.05$), and for the "old" sample it indicated that the value 9.70 is a significant outlier ($P < 0.05$). If these 2 outliers are censored, the difference between young and old is still significantly different ($P < 0.0001$).

Methods: the text should be reviewed to make abbreviations homogeneous (hours appear both as h and hr, days is sometimes abbreviated and sometimes not, milliliters appear both as ml and mL, etc.). Other minor corrections should be that oltripaz appears in capitals in Results, but not in Methods, and the text size in lines 347-351 is different from other parts of the text.

We thank the reviewer for the careful review of the manuscript and have now corrected these discrepancies.

Reviewer #4 (Remarks to the Author):

In this manuscript, Jasper and colleagues reported an interesting phenomenon termed "proteostatic checkpoint".

They showed that intestinal stem cells (ISCs) in *Drosophila* undergo a transient cell cycle arrest upon the induction of protein aggregation. They further demonstrated that the proteostatic checkpoint is regulated by autophagy gene *Atg8a*, Nrf2 homolog *CncC*, and cell cycle inhibitor *Dacapo*. They also found that aggregate-induced cell cycle arrest is lost in old ISCs, while over-expression of *CncC* or *Atg8a* preserved intestinal barrier function in old guts. Overall, the manuscript is well prepared and presented. The finding of “proteostatic checkpoint” and its regulation in ISCs during aging is significant and the study uncovers a novel mechanism underlying the maintenance of tissue homeostasis in ISCs during aging and stress. Although most of the conclusions are well supported, I have a few concerns listed below that need to be addressed before the manuscript is considered for publication.

We would like to thank the reviewer for the positive assessment of our work. We believe that our new data now address most of the remaining concerns:

1. Several methods were used to induce proteostatic stress, such as RNAi against *Rpn3* or *Rnp2*, over-expression of mRFP-Htt[Q138]. It is clear that Htt[Q138]-induced aggregation inhibits cell proliferation (under regenerative pressure), and activates Nrf2 signaling in ISCs. However, to demonstrate that the “proteostatic checkpoint” is not specifically triggered by Htt[Q138], it would be great to show that cell cycle arrest and activation of Nrf2 signaling can also be achieved by silencing proteasome component *Rpn3* or *Rpn2*.

We agree with this point, and had in fact shown in the original manuscript that similar to HttQ138 expression, poly-ubiquitin aggregates induced by *Rpn3* knockdown induce an ISC cell cycle arrest (Fig. 1a-b). We now further show that similar to HttQ138 expression, *Rpn3* knockdown increases Nrf2 activity in ISCs (Fig. 2f), and leads to increased expression of *dacapo* and *Atg8a* in intestinal stem cells (Fig. 4a, c)

2. In Fig. 3, knockdown of *Atg8a* rescued ISC growth, which is probably due to the release of Keap1 and the reduction of Nrf2 activities. Interestingly, *Atg8a* RNAi also impaired aggregate clearance (Fig. 3c) and a slight reduction of ISC growth (Fig. 3d, “+” group). I wonder if *Atg8a* could have two distinct actions on “proteostatic checkpoint”. On the one hand, *Atg8a* knockdown can block autophagosome formation and impair autophagy, which then directly induces protein aggregation and triggers cell cycle arrest. On the other hand, *Atg8a* RNAi can repress Nrf2 via Keap1 and attenuate Htt[Q138]-induced cell cycle arrest. Authors need to address these possibilities in the discussion section.

We thank the reviewer for pointing out these interesting possibilities and we are now discussing this point in detail. We have now also tested whether *Atg8a* and *Atg1* over-expression can suppress HttQ-induced proliferative arrest, and interestingly find that *Atg8a* over-expression has no effect on the cell cycle arrest, while *Atg1* over-expression suppresses HttQ-mediated inhibition of ISC proliferation (Fig. 4d). This further supports the notion that *Atg8* has a dual role in the control of the proteostatic checkpoint and of ISC proteostasis.

To further support the role of *Atg8a* in “proteostatic checkpoint”, it would be great to show if proteostatic stress can induce the expression of *Atg8a*. *Atg8a* reporter (mCherry-*Atg8a*) or *Atg8a* antibody (like GABARAP (E1J4E) antibody from Cell Signaling Technology) can be used.

We have followed this suggestion and indeed now show that accumulation of poly-ubiquitin aggregates induced by *Rpn3* knockdown leads to increased expression of *Atg8a* in intestinal stem cells (using an *Atg8a* antibody generated by the lab of Gabor Juhasz; Fig. 4a)

3. Authors found that the “proteostatic checkpoint” machinery is impaired in old ISCs, while over-expression of *CncC* slows down age-induced intestinal barrier dysfunction. However, it is not clear whether over-expression of *CncC* or Nrf2 activator oltipraz can restore “proteostatic checkpoint” machinery and proteostatic stress-induced cell cycle arrest in old ISCs. Authors need to address this possibility by monitoring the cell proliferation upon Ecc15 treatment or performing lineage-tracing assay in old ISCs expressing both Htt[Q138] and *CncC* (or oltipraz feeding).

We agree with this point and would like to point out that the originally presented data already showed that:

- Oltipraz treatment can increase proteasomal activity in ISCs of old animals
- Oltipraz treatment can improve the clearance of protein aggregates in ISCs of old animals
- *CncC* and *Atg8a* over-expression in ISCs are sufficient to limit barrier dysfunction in old animals.

To further address this question, we have now performed a series of experiments to build on these observations and show now that:

- (i) Oltipraz treatment decreases the age-related accumulation of endogenous Rho1 aggregates in intestinal stem cells (Fig. 6b),
- (ii) Oltipraz limits high proliferation rates of intestinal stem cells in old flies (Fig. 6e),
- (iii) overexpression of CncC and Atg8a in ISCs rescues the age-related accumulation of endogenous poly-ubiquitinated aggregates (Fig. 6c).

We also tested whether Oltipraz treatment reduces intestinal barrier dysfunction in old flies, and indeed observed a reduced frequency of 'smurf' flies, but these data did not reach significance due to technical issues. We include these data in this letter (Fig. R1).

We believe that the combination of these findings indeed support the notion that increasing CncC activity is sufficient to restore the proteostatic checkpoint in ISCs of old flies.

Minor issues:

1. Unlike other figures, Figure 4 used capital letters for each panel, instead of lower-case letters.

We have corrected that.

2. Line 718, remove "e, f"

We have corrected this.

3. How many fly guts were used in each RNA-seq analysis? How many biological replicates per condition?

We now provide this information in the methods section.

4. Authors should include the list of candidate genes identified from their genetic screen on mortality of PQ-treated and Htt[Q138]-expressing flies, unless these data will be used in a different publication.

We are now listing these genes in the text.

5. In some of the experiments, flies were placed at 29 degree (e.g., to induce Htt[Q138]), then back to 18 degree.

There is some concern of the heat shock response triggered by high temperature. Could the heat shock response confound the proteostasis analysis and aggregate-related ISC proliferation and cell cycle arrest?

We have carefully assessed this question, but we believe that this short-term exposure to 29°C does not confound our results, because we do not observe any of the described phenotypes in control wild-type flies exposed to the same temperature shifts. Supporting this assessment are the data shown in Fig. 1d, where HttQ138 expression in ISCs was induced using a drug-inducible system (no temperature shift). Similar to the temperature-inducible system we observed a transient cell cycle arrest when aggregates are present, and a clearance of aggregates in about 2 weeks after induction.

In oltipraz feeding experiment, flies were aged at 18 degree for a long period of time. Because flies age at a slower rate under low temperature. Could these results provide any insights into the regulation of proteostatic checkpoint during normal aging in *Drosophila*?

We agree that this is a consideration. However, we do observe significant age-associated phenotypes (over-proliferation, dysplasia, and barrier dysfunction) also in flies aged at the lower temperature. In general, aging phenotypes are simply delayed at 18°C compared to 25°C in flies.

Rodriguez-Fernandez_FIGURE R1

Figure R1: Evidence that Oltipraz exposure reduces the incidence of barrier dysfunction in old flies. Compare to Fig. 6f.

Flies were aged to 30 days at 18°C on standard food, then were fed food supplemented with 500 μ M Oltipraz with 0.1% DMSO or with 0.1% DMSO (mock) for 15 days, exposed to 29°C for 24 hours to induced transgene expression (to maintain conditions similar to Fig. 6f) and then tested for barrier function by placing them on 'Smurf food' (i.e. regular fly food mixed with FD&C blue dye #1 a non-absorbable food dye) for 10 days. The percent of blue flies in each cohort (cohort sizes were $n = 14, 15$ female flies for Mock treatment, and $n = 14, 14, 11$ female flies for Oltipraz treatment) was assessed at 55 days of age. Unpaired two-tailed t-test: ns, not significant. Genotype: *esgGal4, tub::Gal80^{ts}, Su(H)::Gal80/UAS-luciferase*.

Reviewers' comments:

Reviewer #1 (Remarks to the Author):

As noted in my previous review, this is a very original paper with an extensive array of striking results and an interesting, significant message. The authors have addressed many of the reviewers' comments appropriately; the paper has been edited to fix over-stated conclusions and unclear issues, and a modest amount of new data added. Overall it's a nice work. However a number of my previous concerns were not carefully addressed, even though this should have been rather straightforward. These are listed below. Some (#2,4) have significant bearing on the paper's impact.

1. Previous Question 1. The authors still refer to numbers and amounts of protein "aggregates" at many points in the paper, even though they never quantified these. Instead, they quantified total RFP signals in all examples except Fig 6d. Yes, we can see aggregates in the pictures, but just a few, in just a few cells, and pictures like this are not sufficient to document the process convincingly. The fly gut is notoriously variable when stress-dependent phenotypes are assayed. Either the aggregates should be quantified somehow, or the text should be re-worded to reflect what was actually measured (total RFP signal).

2. Fig 3e in the revision contains a requested experiment using dacapo-RNAi to confirm the marginally convincing data using *dap/+* heterozygote mutants (previous question #7). Indeed the effect on RFP signal (protein "aggregates") is as predicted. However the Figure lacks a critical panel, namely data showing the effect of dacapo-RNAi on the proteostasis checkpoint cell cycle arrest. The expectation is that suppressing *dap* will release the cell cycle arrest, and indeed an important conclusion of the paper is that *dap* induction is what mediates that arrest. The data in the paper does not yet strongly support that conclusion, so the missing experiment with *Dap*-RNAi is quite important.

3. The data in Fig 5a is not convincing.

4. As noted before, although the paper is ostensibly about aging, there is still very little data on how this new proteostasis checkpoint affects functional decline in the gut, and none on lifespan. The authors' response to this (previous question #12) is essentially empty text.

5. The Oltipraz results in fig 6b,e require more controls. These effect could just reflect Oltipraz suppressing ISC division, or stemness.

6. The UPRER is still not discussed at length (previous question #13), though the earlier publication on that topic is now cited. A bit more could be said about the previous work, to put this new report in context.

Reviewer #3 (Remarks to the Author):

The authors adequately carried out the requested experiments and addressed all concerns raised in previous review round.

I think that this revised manuscript is suitable for publication on Nature Communications.

Reviewer #4 (Remarks to the Author):

Authors have addressed most of my previous comments. The revised manuscript has met the publication criterion. I recommend it to be accepted for publication in Nature Communications.

Response to reviewers NCOMMS-17-26472A

We would like to thank the three reviewers again for their positive reception of our work. Reviewers #3 and #4 both found our response to their initial review to satisfy their concerns and recommended publication. Reviewer #1 also found that our paper is 'very original' and provides an 'extensive array of striking results' and has 'an interesting, significant message'. While reviewer #1 also found that we had addressed many comments appropriately and had improved the manuscript, there were a few instances where additional clarification was needed. We have now performed additional experiments to address these remaining concerns and have edited the manuscript to clarify any lingering issues. We believe that the manuscript has significantly improved as a result and we hope the reviewer now agrees that the reported findings are of sufficient interest to be published in *Nature Communications*.

Reviewers' comments:

Reviewer #1 (Remarks to the Author):

As noted in my previous review, this is a very original paper with an extensive array of striking results and an interesting, significant message. The authors have addressed many of the reviewers' comments appropriately; the paper has been edited to fix over-stated conclusions and unclear issues, and a modest amount of new data added. Overall it's a nice work. However a number of my previous concerns were not carefully addressed, even though this should have been rather straightforward. These are listed below. Some (#2,4) have significant bearing on the paper's impact.

We again thank the reviewer for the positive reception of our work, and apologize if we did not address all concerns appropriately in our first revision. We have now performed additional experiments to address these concerns and have edited the manuscript as requested. We explain these changes in detail below:

1. Previous Question 1. The authors still refer to numbers and amounts of protein "aggregates" at many points in the paper, even though they never quantified these. Instead, they quantified total RFP signals in all examples except Fig 6d. Yes, we can see aggregates in the pictures, but just a few, in just a few cells, and pictures like this are not sufficient to document the process convincingly. The fly gut is notoriously variable when stress-dependent phenotypes are assayed. Either the aggregates should be quantified somehow, or the text should be re-worded to reflect what was actually measured (total RFP signal).

We understand this concern and had changed the text in several instances to address it. But we did not sufficiently clarify our reasons for taking total RFP signal as a proxy for aggregates:

When evaluating the presence or absence of RFP-Htt^{Q138} aggregates in ISCs, we observe aggregation of the RFP-Htt^{Q138} protein invariably after the first 24 hours of expression, and when wild-type ISCs are evaluated 2 weeks later, these aggregates (and hence the RFP signal) are gone. However, aggregate size and aggregate number/cell can vary substantially even 24 hours after induction of the transgene. In order to obtain a robust measure for RFP-Htt^{Q138} aggregates present in ISCs, we therefore opted to quantify total RFP signal. Since RFP signal is only observed in the form of aggregates in these conditions, we believe that it represents an unbiased metric for aggregate presence in ISCs.

We therefore used 'aggregates' rather than 'RFP signal' in various instances in the text. But to clarify this issue further, we have now edited the text to (i) explain more clearly our reason to use RFP signal as a proxy for aggregates, and (ii) refer more precisely to the measured signal where appropriate. Changes in the text are tracked.

2. Fig 3e in the revision contains a requested experiment using dacapo-RNAi to confirm the marginally convincing data using dap/+ heterozygote mutants (previous question #7). Indeed the effect on RFP signal (protein "aggregates") is as predicted. However the Figure lacks a critical panel, namely data showing the effect of dacapo-RNAi on the proteostasis checkpoint cell cycle arrest. The expectation is that suppressing dap will release the cell cycle arrest, and indeed an important conclusion of the paper is that dap induction is what mediates that arrest. The data in the paper does not yet strongly support that conclusion, so the missing experiment with Dap-RNAi is quite important.

We agree that the dapRNAi experiments are important, although we disagree that the dap/+ heterozygote results were marginally convincing (effect sizes were strong and the data were statistically significant; furthermore, haploinsufficiency is not surprising for a gene that needs to be rapidly induced to elicit an effect).

As requested, we have now performed additional experiments that we believe further strengthen the evidence for a role for Dacapo as a critical effector of the proteostasis checkpoint:

- We show that knockdown of *dap* with *dap*RNAi prevents the inhibition of *Ecc15*-induced ISC proliferation by RFP-Htt^{Q138} expression (new panel in Fig 3h), i.e. releasing the aggregate-induced cell cycle arrest.
- We show that Paraquat – induced mortality in animals in which RFP-Htt^{Q138} is expressed in ISCs is reduced after *dap*RNAi expression, recapitulating the results with *dap* heterozygotes (Fig. 3c).
- we now include new data using fluorescent *in situ* hybridization that show that *dap* is induced transcriptionally downstream of *CncC* (Fig. 4d; previous data only included antibody staining).

3. The data in Fig 5a is not convincing.

It is not entirely clear to us why these data are not convincing, but we have added additional representative images to make sure we properly illustrate the point, namely that ISCs in old flies exhibit Rho1 aggregates.

4. As noted before, although the paper is ostensibly about aging, there is still very little data on how this new proteostasis checkpoint affects functional decline in the gut, and none on lifespan. The authors' response to this (previous question #12) is essentially empty text.

We apologize if our response to previous point #12 was not adequate. The answer was:

'We now show more extensively that endogenous protein aggregates accumulate in ISCs of aging flies, and that this can be prevented by restoration of *CncC* activity (Figs. 5 and 6). Together with our demonstration that *CncC* and *Atg8a* over-expression prevent barrier dysfunction in old flies (Fig. 6e), we believe that these data in fact show that the proteostasis checkpoint is an essential promoter of epithelial homeostasis.'

We would like to explain further why we believe that our data in fact support the idea that the proteostasis checkpoint is needed to maintain tissue homeostasis in the intestinal epithelium of aging flies:

Our data show that intestinal stem cells of aging flies exhibit a breakdown of proteostasis that includes the increased presence of protein aggregates (Fig. 5a) concomitant with a decline of proteasome activity (Fig. 5b). We further show that the clearance of induced protein aggregates is impaired in ISCs of old flies (Fig. 5c), and that this is due to the loss of the proteostasis checkpoint in these ISCs (Fig. 5d, e, f). We further show that treatment with Oltipraz to activate *CncC* can restore the proteostasis checkpoint in old flies (Fig. 6a), thus reducing the amount of endogenous protein aggregates observed in old ISCs (Fig. 6b) and restoring clearance of Htt^{Q138} aggregates (Fig. 6d). This is consistent with the activation of *CncC* by Oltipraz, as *CncC* over-expression is sufficient to promote aggregate clearance in old ISCs (Fig. 6c).

We had previously shown that *CncC* over-expression in ISCs is sufficient to preserve epithelial homeostasis in aging intestines (Hochmuth et al., 2011), and epithelial homeostasis is known to influence lifespan (Biteau, Karpac, et al., 2010, Rera et al., 2012). Our manuscript added new insight to this body of work and linked it to the restoration of the proteostasis checkpoint by (i) showing that Oltipraz treatment can restore epithelial homeostasis (reducing excessive ISC proliferation, Fig. 6e), and (ii) showing that *CncC* and *Atg8a* over-expression in ISCs is sufficient to limit epithelial barrier dysfunction in aging flies (Fig. 6f).

In response to the reviewer's comments, we have now performed additional experiments to test whether Oltipraz treatment is also sufficient to extend lifespan and limit barrier dysfunction in aging flies. Continuous administration of Oltipraz throughout the lifespan of the animal does not lead to longevity (Dirk Bohmann, personal communication), and we therefore tested whether intermittent exposure to Oltipraz would influence lifespan. Strikingly, lifespan is in fact significantly extended in both wild-type and RFP-Htt^{Q138} expressing animals when they are exposed to Oltipraz intermittently at older ages (for 15 days at 30 days of age and then again for 4 days at 56 days of age; Fig. 6g). This lifespan extension is accompanied by a delayed loss of epithelial barrier function (Fig. 6h), indicating that stimulation of the proteostatic checkpoint in ISCs of old flies promotes longevity by enhancing epithelial homeostasis.

We believe that in sum these data provide significant evidence for one of the conclusions of the paper, namely that loss of the proteostasis checkpoint contributes to the age-related loss of epithelial function. It should also be noted, however, that this is only one of the conclusions of the paper, as arguably the more critical finding is the identification and characterization of the proteostasis checkpoint itself. We have now edited the abstract of the manuscript to better balance our message.

5. The Oltipraz results in fig 6b,e require more controls. These effect could just reflect Oltipraz suppressing ISC division, or stemness.

We agree, and apologize for the omission, we have now included the respective control in Fig. 6e, showing that Oltipraz does not affect ISC proliferation induced by Ecc15 infection in a wild-type background.

6. The UPRER is still not discussed at length (previous question #13), though the earlier publication on that topic is now cited. A bit more could be said about the previous work, to put this new report in context.

We had in fact added a paragraph on the UPRER to the discussion:

The activation of CncC after cytosolic proteostatic stress differs mechanistically and in its consequence from the regulation of CncC after other types of protein stress in ISCs: in response to unfolded protein stress in the ER, CncC is specifically inactivated by a ROS/JNK-mediated signaling pathway^{24,28}. This mechanism allows ISC proliferation to be increased in response to tissue damage, but can also contribute to the loss of tissue homeostasis in aging conditions⁶⁴. The activation of CncC after cytosolic protein stress, in turn, allows arresting ISC proliferation during protein aggregate clearance. Our data suggest that this is achieved by coordinated induction of the cell cycle inhibitor Dacapo and a battery of genes encoding proteins involved in proteolysis.

Evidently, the reviewer was expecting a more expansive discussion of the UPR-ER, and, while we are under strict length limits for this manuscript, we have now expanded on that topic, both in the introduction and the discussion. Changes to the text are tracked.

Reviewer #3 (Remarks to the Author):

The authors adequately carried out the requested experiments and addressed all concerns raised in previous review round.

I think that this revised manuscript is suitable for publication on Nature Communications.

We thank the reviewer for the positive reception of our work.

Reviewer #4 (Remarks to the Author):

Authors have addressed most of my previous comments. The revised manuscript has met the publication criterion. I recommend it to be accepted for publication in Nature Communications.

We thank the reviewer for the positive reception of our work.

REVIEWERS' COMMENTS:

Reviewer #1 (Remarks to the Author):

We'd like to thank the authors for their patience in waiting for this review. Overall, this second set of revisions is quite satisfactory. The authors have substantially addressed our previous comments. The experiments in Fig.3h (Dap-RNAi) and Fig.4d (Dap in situ) were nicely carried out, and the results are consistent with the with the authors' conclusions. The new lifespan data shown in Fig. 6gh is also an important addition that substantiates their proposals about the proteostasis checkpoint and it's function in ISCs. The revised manuscript has met reasonable publication criterion, and we look forward to seeing it online.

Response to reviewers NCOMMS-17-26472C

We would like to thank the three reviewers again for their positive reception of our work. We believe that the manuscript has significantly improved as a result of the reviewers' suggestions, and we are glad that reviewer #1 now agrees that the reported findings are of sufficient interest to be published in *Nature Communications*.

Reviewers' comments:

Reviewer #1 (Remarks to the Author):

We'd like to thank the authors for their patience in waiting for this review. Overall, this second set of revisions is quite satisfactory. The authors have substantially addressed our previous comments. The experiments in Fig.3h (Dap-RNAi) and Fig.4d (Dap in situ) were nicely carried out, and the results are consistent with the authors' conclusions. The new lifespan data shown in Fig. 6gh is also an important addition that substantiates their proposals about the proteostasis checkpoint and its function in ISCs. The revised manuscript has met reasonable publication criterion, and we look forward to seeing it online.

We thank the reviewer for the positive reception of our work.